

# Cloud-based real-time enhancement for disease prediction using Confluent Cloud, Apache Kafka, feature optimization, and explainable artificial intelligence

Abdulaziz AlMohimeed

College of Computer and Information Sciences, Imam Mohammad Ibn Saud Islamic University (IMSIU), Riyadh, Saudi Arabia

## ABSTRACT

In recent years, Internet of Things (IoT)-based technologies have advanced healthcare by facilitating the development of monitoring systems, subsequently generating an exponential amount of streaming data. This streaming data can be preprocessed and analyzed using technologies that integrate ensemble models, Explainable Artificial Intelligence (XAI), feature selection (FS) method and big data streaming processing platforms to develop predictive real-time systems. This integration adds new value to healthcare that helps organizations enhance clinical decision-making, improve patient care, and elevate the overall quality of healthcare. This article presents a real-time system for the early detection and treatment of chronic kidney disease (CKD) using a real-world simulation application. The real-time system is developed in two phases. The first phase aims to propose a stacking model, apply a genetic algorithm (GA) and Particle swarm optimization (PSO) as feature selection, and explore a stacking model with the best features with explainable artificial intelligence (XAI). The best model with the best-optimized features is used to develop the second phase. The results showed that stacking model with GA is achieved the hightest performance with 100 accuracy, 100 precision, 100 recall, and 100 F1-score. The second phase is designed based on Confluent Cloud, which offers several benefits for creating a real-time streaming system based on Apache Kafka, providing multiple APIs—the Producer API and Consumer API—for data producers and consumers, respectively. Python scripts are developed to pipeline streaming data. The first Python script to generate streaming health attributes that are pushed into a Kafka topic. A second Python script to consume health attributes from a Kafka topic and apply a stacking model to predict CKD in real-time. The results showed that the stacking model with features selected by GA recorded the best performance with 100 accuracy. The pipeline's streaming steps have validated our approach's effectiveness in real-time, leveraging Confluent Cloud and Apache Kafka.

Corresponding author
Abdulaziz AlMohimeed,
aialmohimeed@imamu.edu.sa

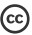

## INTRODUCTION

Chronic kidney disease (CKD), characterized by the kidneys' inability to remove waste and fluids while maintaining essential blood balances, poses severe health risks due to the accumulation of debris and fluids in the body (*Romagnani et al., 2017*; *Stenvinkel, 2010*). This condition affects electrical balance, hormone secretion, and blood pressure regulation. Specialized medical attention is crucial for treating chronic kidney failure, often involving renal function replacement techniques like artificial dialysis or kidney transplants (*Kemph, 1966*; *Nugent et al., 2011*; *Luyckx, Tonelli & Stanifer, 2018*). The integration of machine learning (ML), ensemble learning, feature selection methods, and explainable artificial intelligence (XAI) has played an essential role in improving diagnosis, treatment, and patient care. ML models are developed using clinical data and laboratory analyses and offer a powerful tool for identifying patterns and signs of chronic kidney failure, allowing early diagnosis before apparent symptoms appear (*Cao & Pan, 2024*; *Xue et al., 2023*). In addition, ensemble learning leveraging the collective intelligence of multiple ML algorithms has emerged as a recent advancement to mitigate biases or errors in individual models (*Rokach, 2010*; *Zhou & Zhou, 2021*; *Luo et al., 2024*). Stacking is a type of ensemble learning that is designed to integrate the output of various individual models to train and evaluate a meta-model that capitalizes on the collective knowledge of individual base models, offering improved performance and robustness (*Wolpert, 1992*). Furthermore, feature selection is a crucial step in model development, aiming to reduce the number of features and enhance performance. Techniques like genetic algorithms (GA) and particle swarm optimization (PSO) are employed for this purpose. Additionally, we delve into stacking models with XAI to identify the most effective approach.

Recently, digital data has become increasingly important in various domains of life, including healthcare, science, technology, and society. Multiple sources, such as sensors and mobile applications, have been capturing and generating a large amount of data, collectively referred to as big data. Streaming data, characterized by a continuous stream of information, requires real-time processing and transmission (*Kitchin, 2014*). A streaming data scenario enables the generation of real-time insights and decisions based on the processing and analysis of information as it is produced (*Szewczyk, 2011*). Managing vast and streaming datasets poses challenges in storage, processing, visualization, and knowledge extraction. Big data analytics and artificial intelligence are pivotal in addressing these challenges, playing crucial roles in real-time healthcare applications. They empower systems to make intelligent predictions and decisions in near real-time, enhancing the overall efficiency of healthcare processes. Several modern stream processing architectures are used for real-time data analytics, such as Apache Kafka (*Garg, 2013*) and Apache Spark (*Salloum et al., 2016*). These architectures are implemented based on Cloud environments such as the Confluent platform (*Confluent, 2014*) and Microsoft Azure (*Copeland et al., 2015*) for Internet of Things (IoT) technologies.

### Motivations and contributions

We introduce a novel framework to address critical gaps in previous studies identified by the role of integrating stacking models, genetic algorithms as feature selection, XAI, and

streaming data platforms to efficiently analyze and process health streaming data, enabling the early prediction of CK disease in real-time. Previous studies (*Chittora et al., 2021*; *Qin et al., 2019*; *Gunarathne, Perera & Kahandawaarachchi, 2017*; *Dritsas & Trigka, 2022*; *Iftikhar et al., 2023*; *Song et al., 2020*) applied ML models such as LR, SVM RF, KNN, SVM and XGBoost to predict CKD but they do not integrating AI models with big data streaming to predict CKD in real-time. Other studies (*Chittora et al., 2021*; *Polat, Danaei Mehr & Cetin, 2017*; *Almasoud & Ward, 2019*) apply base features selection methods such as correlation-based, LightGBM to select the most important features from CKD dataset but do not apply features selection methods based on GA or PSO. Furthermore, existing studies do not utilize XAI, stacking models, or the integration of stacking models with XAI in real-time systems. Additionally, studies such as *Ed-daoudy & Maalmi (2019)*, *Ed-daoudy, Maalmi & El Ouaazizi (2023)*, and *Saleh et al. (2021)* have developed real-time medical systems using big data platforms; however, they lack real-world screening simulations based on Confluent Cloud.

To address this gap, we explore the role of the stacking model and GA feature selection in improving predictive capabilities in healthcare care, interpret the stacking model's result with XAI, and emphasize the importance of stream processing platforms for developing stream-streaming pipeline solutions based on Confluent Cloud for cloud-based data processing, Apache Kafka for real-time data streaming. The focus is on introducing real-time solutions for the early detection and treatment of CKD.

Summary of this article's contributions:

- Developing a real-time system that integrates stacking models, GA as feature selection methods, XAI, and big data streaming (Confluent Cloud and Apache Kafka) to predict CKD in real-time and improve the healthcare monitoring system.
- Investigating the impact of both individual ML models and stacking ML models on selected features by GA and PSO to predict CKD. Through comprehensive performance using different evaluation metrics, accuracy precision, recall, and F1-score.
- Incorporating explanation tools alongside predictive capabilities provides transparent and understandable insights into the stacking model. This enhances interpretability and aids healthcare professionals in understanding the underlying factors that contribute to the prediction of CKD.
- Developing streaming real-time pipelines using Python scripts to generate streaming health attributes and push them to a Kafka topic. Another consumes health attributes from a Kafka topic and applies the best model (stacking model) for real-time prediction of CKD by leveraging Confluent Cloud and Apache Kafka.

## Article structure

According to the structure of the study: In "Literature Review", the related works are presented. "Methodology" describes the two phases of developing our work. "Results" presents the results of our experiments. "Discussion" provides a discussion of the work. Finally, in "Conclusion", the study is concluded.

## LITERATURE REVIEW

This section will present studies and research efforts that have focused on real-time applications and predicting CKD.

### Predicting CKD using ML/DL models

Studies have applied ML models with feature selection to predict CKD (*Alshanbari et al., 2023*; *Cuba et al., 2024*). *Chittora et al. (2021)* registered the results of seven ML models applied to full features and selected. Correlation-based and Wrapper methods were used as feature selection methods. The results showed that SVM recorded the highest accuracy. *Qin et al. (2019)* proposed a hybrid model that integrated logistic regression (LR) with Random Forest (RF). The hybrid model recorded the highest accuracy compared to LR, RF, support vector machine (SVM), K-nearest neighbor (KNN), naive Bayes (NB), and feed-forward neural networks. *De Almeida et al. (2020)* applied RF, SVM, and decision tree (DT) to the dataset extracted from MIMIC-II to predict CKD. The results concluded that RF and DT got the highest results. *Gunarathne, Perera & Kahandawaarachchi (2017)* proposed a multiclass decision forest algorithm compared to multiclass decision jungle, multiclass logistic regression, and multiclass neural network. The results showed that their proposed model recorded the highest accuracy. *Polat, Danaei Mehr & Cetin (2017)* applied SVM with two feature selection (FS) methods: Wrapper (first search engine) and filter (correlation). The results showed that SVM with a filtered method recorded the highest accuracy. *Almasoud & Ward (2019)* applied four ML classifiers: LR, SVM, RF and gradient boosting, and A filter FS method was applied to reduce the number of features. The gradient boosting algorithm achieved the highest accuracy. *Xiao et al. (2019)* compared several predictive models using statistical, ML and neural network approaches such as XGBoost, LR, LASSO regression, SVM, RF, ridge regression, and KNN using 551 patients with CKD from the Department of Nephrology, Huadong Hospital, Shanghai Fudan University Affiliated Hospital. *Iftikhar et al. (2023)* predicted CKD using different ML models, including LR, RF, DT, KNN, and SVM using the dataset collected from a case-control study containing CKD patients from district Buner, Khyber Paktunkhwa, Pakistan. The results confirmed that SVM recorded the best performance compared to other models. *Tazin, Sabab & Chowdhury (2016)* applied SVM, DT, NB, and KNN to detect CKD. DT recorded the highest accuracy. *Krishnamurthy et al. (2021)* applied the LightGBM model to select the essential features. They used a convolutional neural network (CNN) to predict CKD. CNN achieved the best performance compared to other models. *Segal et al. (2020)* used XGBoost for the early diagnosis of renal illness. Extreme gradient boosting (XGBoost) was compared against RF and CatBoost, which recorded the best performance.

Table 1 presents the summary of related works.

### Real-time systems for medical applications

Several studies and research efforts have focused on real-time applications. In *Ed-daoudy & Maalmi (2019)*, the authors develop a DT model based on SparkML to predict heart and diabetes diseases. They integrated spark streaming, Kafka, and DT to develop a real-time

**Table 1 Summary of related works.**

| Articles | Models | Feature selection (FS) | Limitation |
|---|---|---|---|
| *Chittora et al. (2021)* | ML models | Correlation-based and Wrapper methods | Just apply ML with FS |
| *Qin et al. (2019)* | Hybrid model | No | Just apply ML |
| *De Almeida et al. (2020)* | ML models | No | Just apply ML |
| *Gunarathne, Perera & Kahandawaarachchi (2017)* | ML models | No | Just apply ML |
| *Polat, Danaei Mehr & Cetin (2017)* | SVM | Filter and Wrapper methods | Just apply ML with FS |
| *Almasoud & Ward (2019)* | ML models | No | Just apply ML |
| *Xiao et al. (2019)* | ML and DL models | No | Just apply ML |
| *Iftikhar et al. (2023)* | ML models | No | Just apply ML |
| *Tazin, Sabab & Chowdhury (2016)* | ML models | No | Just apply ML |
| *Krishnamurthy et al. (2021)* | CNN | LightGBM | Just apply ML with FS |
| *Segal et al. (2020)* | XGBoost, RF and CatBoost | No | Just apply ML |

system that could predict heart and diabetes. In *Ed-daoudy, Maalmi & El Ouaazizi (2023)*, the authors proposed a real-time system based on Twitter, Kafka, Spark, and ML to predict heart disease. They applied various feature selection methods to improve the performance of the model. The results showed that FS with ML recorded the best performance. In *Ahmed et al. (2020)*, the authors applied SparkML and Spark Streaming to develop a real-time system. They applied FS methods with DT SVM—RF and LR to get the best real-time model to predict heart disease. In *Saleh et al. (2021)*, the author applied different DL models to the historical time series blood pressure dataset to forecast BP. The best DL model was integrated with The simulated sensor, Kafka, and Spark streaming to forecast BP in near real-time. In *Karim, Sahay & Rebholz-Schuhmann (2015)*, the authors proposed a framework for real-time healthcare monitoring for public health surveillance. SparkML was utilized to develop ML models: SVM, RF, and K-Means clustering libraries. Spark streaming was used to pre-processing and receive data from Wearable sensors and Internet of Things (IoT) devices. *Tun & Khine (2020)* suggested a cardiac diagnosis analysis system based on Apache Spark that uses a DL pipeline to categorize electrocardiogram (ECG) images. The proposed system detected cardiac disorders by training ECG segmented images, eliminating redundant segmented images using principal component analysis (PCA) and transfer learning, and deep learning pipeline classification.

# METHODOLOGY

We propose a significant enhancement in healthcare by integrating machine learning (ML) models, streaming data, and big data streaming platforms. This study pioneers the development of a real-time prediction system that seamlessly integrates both offline and online phases, as illustrated in Fig. 1. The primary objective of the offline model phase is to determine the optimal models with the most effective features to be utilized in the subsequent online phase. The online phase is constructed around Python applications (producer and consumer) seamlessly integrated with Apache Kafka on the Confluent platform. Each phase is elaborated upon in detail below.

Data sources

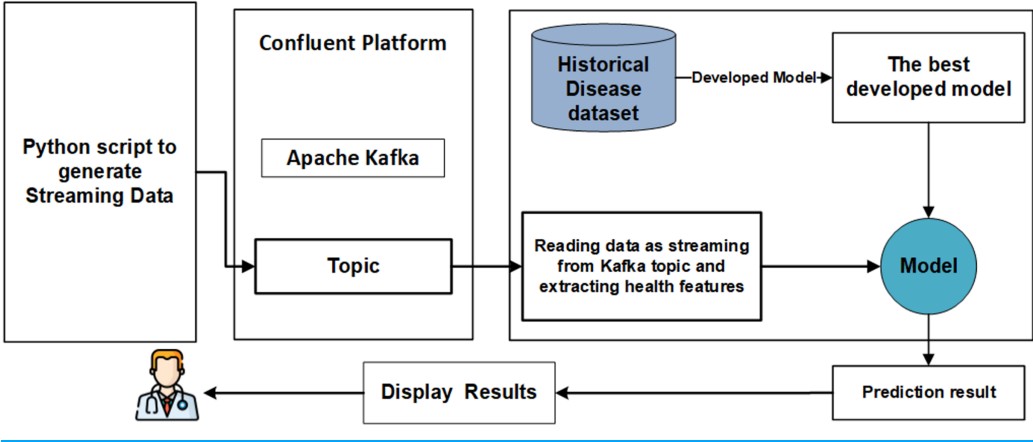

**Figure 1 The main phases of real-time system.**

## Offline model phase

The objective of this section is to introduce a stacking model that incorporates feature selection methods using genetic and PSO techniques to predict CKD. Illustrated in Fig. 2, the offline phases encompass various stages, namely data collection, preprocessing, FS methods, baseline ML, and stacking models. The subsequent sections provide a detailed overview of each stage.

### Data description

This study utilized UCI's machine learning repository to aggregate the baseline data set on chronic kidney disease (*Rubini, Soundarapandian & Eswaran, 2015*). The data set includes 400 cases of CKD, 150 of which are negative and 250 of which are positive. The data set is composed of 24 features that are classified into 13 categorical features and 11 numeric features, with a class label that has two values: 1 and 0. The details of the characteristics of each feature are shown in Table 2.

### Data preprocessing

We applied various data preprocessing steps:

- Data encoding: the dataset includes categorical and numeric features. The Scikit-learn library is used to encode all categorical features into numerical data.
- Missing values can be filled in with several statistical methods, depending on how much data is missing and how important the missing feature. Mean, maximum, and mode are good statistical techniques when 5% to 10% of values are missing. Our study has a low number of missing values, which is handled using mean.

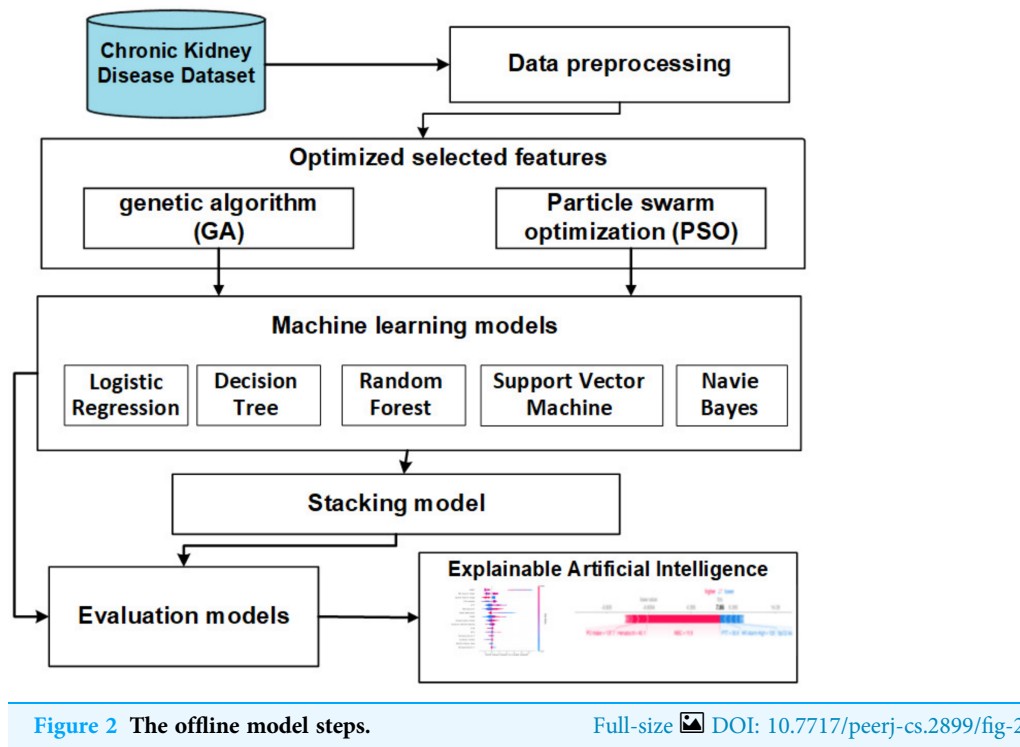

**Figure 2  The offline model steps.**     

### Splitting datasets

The dataset was split into different cases and we performed experiments in different cases, including Experimental 1 (80% training set and 20% testing set), Experimental 2 (70% training set and 30% testing set) and Experimental 3 (60% training set and 40% testing set).

### Feature selection methods

Feature selection can be characterized as selecting a smaller selection of appropriate characteristics (variables, attributes) from a more excellent range of available features (*Sánchez-Maroño, Alonso-Betanzos & Tombilla-Sanromán, 2007*; *Jović, Brkić & Bogunović, 2015*). Deleting redundant features reduces the dimensionality of machine learning models and increases performance and efficiency. PSO and GA are feature selection methods used. The advantages of applying GA and PSO are: GA acquired from its characteristics, which gave it the strength and efficiency necessary to be effective in finding optimal local solutions (*Alam et al., 2020*). Also, its flexibility gave it the ability to adapt to the various problems that it may encounter while selecting different features, especially when dealing with big data environments. Its characteristics gave it the power to choose the best features based on the extent of their contribution to the predictive distribution clearly and distinctly. Therefore, it combines exploration to find good solutions with the ability to improve existing solutions effectively (*Alam et al., 2020*). PSO is characterized by ease of implementation due to the limited number of its tuning parameters compared to similar algorithms (*Houssein et al., 2021*). This will lead to its remarkable speed in reaching good solutions due to its inherited effectiveness from its characteristics in finding a rapid convergence to the desired solution. Therefore, although there are many algorithms

**Table 2 Database description.**

| # | Column name | Abb | D.T | Range |
|---|---|---|---|---|
| 1 | Age | Age | N | (2 to 90) |
| 2 | Blood pressure | PB | N | (50 to 180) |
| 3 | Specific gravity | SG | C | (1.025, 10.20, 1.015, 1.010, 1.005) |
| 4 | Albumin | AL | C | (0, 1, 2, 3, 4, 5) |
| 5 | Sugar | SU | C | (0, 1, 2, 3, 4, 5) |
| 6 | Red blood cells | RBC | C | (Abnormal, Normal) |
| 7 | Pus cell | PC | C | (Abnormal, Normal) |
| 8 | Pus cell clumps | PCC | C | (Abnormal, Normal) |
| 9 | Bacteria | BA | C | (Present, Not present) |
| 10 | Blood glucose random | BGR | N | (22 to 490) |
| 11 | Blood urea | BU | N | (1.5 to 391) |
| 12 | Serum creatine | SC | N | (0.4, 76) |
| 13 | Sodium | SOD | N | (4.5 to 163) |
| 14 | Potassium | POT | N | (2.5 to 47) |
| 15 | Hemoglobin | HEMO | N | (3.1 to 17.8) |
| 16 | Packed cell volume | PCV | N | (9 to 43) |
| 17 | White blood cell count | WC | N | (2,200 to 4,800) |
| 18 | Red blood cell count | RC | N | (2.1 to 8) |
| 19 | Hypertension | HTN | C | (Yes, No) |
| 20 | Diabetes mellitus | DM | C | (Yes, No) |
| 21 | Coronary artery diseases | CAD | C | (Yes, No) |
| 22 | Appetite | APPET | C | (Good, Poor) |
| 23 | Pedal edema | PE | C | (Yes, No) |
| 24 | Anemia | ANE | C | (Yes, No) |
| 25 | Class | Class | C | (CKD, Not CKD) |

available for feature exploration, both PSO and GA have shown speed and efficiency in reaching rapid convergence of the available and optimal solutions in feature sets (*Yadav & Anubhav, 2020*).

- PSO is a bio-inspired search and optimization system that uses a population-based strategy to find the best solution for a given problem (*Xue, Zhang & Browne, 2012*). In PSO, a population of potential solutions, known as particles, traverses the search space in search of the optimal or near-optimal solution. Each particle represents a potential solution and progresses through the search space based on its experience and the experience of the best performing particles in the population (*Tran, Xue & Zhang, 2018*). The process begins by populating the search space with randomly distributed particles. Every particle has a position as well as a velocity. The position represents a potential solution, while the velocity specifies the direction and speed of the particle's movement (*Marini & Walczak, 2015*). The particles modify their positions and velocities during each generation or iteration of the algorithm, which is an iteration based on their

individual experiences and the experiences of the top-performing particles in the population.

- GA is a computational tool for addressing difficult search and optimization tasks and extracting or choosing relevant features (*Mitchell, 1998*; *Kramer & Kramer, 2017*). GA iteratively builds a population of chromosomes (solutions) and their genes through selection, crossover, and mutation, with the fittest solutions prevailing. Individuals in the population are evaluated using an objective function or heuristic, which is utilized to choose individuals to reproduce on each repeat (*Forrest, 1996*; *Wright, 1991*). Those samples that perform better in the objective function will have a higher chance of reproduction. The best individual, picked after many generations, is the end result (*Wright, 1991*). GA encodes the optimization function as bit arrays that resemble chromosomes, and Genetic operators customize strings to discover a near-optimal solution to the problem at hand. This is accomplished by following the methods illustrated in Fig. 3 (*Fong et al., 2014*):

  1. Coding the objectives or cost functions.
  2. Creating a fitness function.
  3. A generation is the process of producing a person (solution).
  4. Evaluating the fitness function of individuals in the population.
  5. Creating a new population by crossover and mutation, fitness-proportionate growth, and then replacing the previous population and looping with the new population.
  6. Decoding the outputs to resolve the inquiry.

Crossover, mutation, and selection are the three fundamental genetic operators in genetic algorithms (*Al-Asasfeh, Hamdan & Abo-Hammour, 2013*):

  – Crossover: The swapping of chromosomes or other solutions representing sections of a solution. The primary goal is to give subspace convergence and solution mixing.
  – Mutation: The unplanned alteration of the constituent parts of a single solution enhances population variety and provides a technique to avoid a local optimum.
  – Selection of the fittest is passing on options with high fitness to future generations, usually by selecting the most acceptable solutions.

### Baseline machine learning models

Different ML models are applied:

- Support vector machine (SVM) is a supervised learning technique that is most commonly used for classification tasks. To categorize data points, the algorithm creates a hyperplane or a series of hyperplanes in a high-dimensional feature space (*Wang, 2005*). The objective of SVM is to maximize the margin, which is the distance between the hyperplane and the nearest data points in each class. It can handle both linearly and non-linearly separable data by employing various kernels or mapping functions (*Meyer & Wien, 2001*).

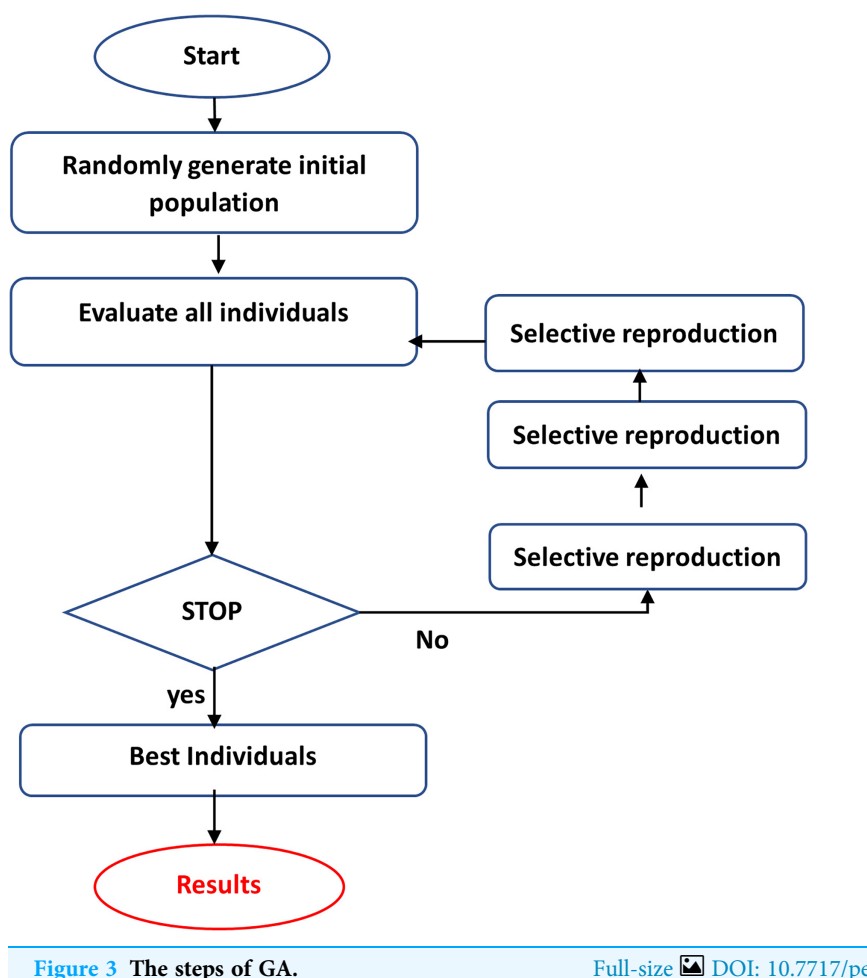

**Figure 3** The steps of GA.  

- Naive Bayes (NB) algorithm (*Webb, Keogh & Miikkulainen, 2010*) is a probabilistic machine learning technique commonly used for classification tasks. NB is founded on Bayes' theorem, which describes the likelihood of an event given prior knowledge or evidence. The assumption of feature independence is made by Naive Bayes, which means that the presence or absence of one feature is unrelated to the presence or absence of any other feature (*Leung, 2007*).

- Decision tree (DT) is a supervised learning algorithm that is adaptable and interpretable. DT can be utilized for classification and regression tasks to create a tree-like structure by recursively splitting the data based on distinct attributes and their values (*Kotsiantis, 2013*). Each internal node corresponds to a feature test, and each leaf node corresponds to a class label or a predicted value. Decision trees can capture complicated linkages and interactions between elements (*Kingsford & Salzberg, 2008*).

- Random Forest (RF) (*Rigatti, 2017*) is a collaborative learning method that makes predictions by combining numerous decision trees. It builds an ensemble of decision trees by training each tree on a random subset of the training data and characteristics.

During prediction, each tree in the forest predicts the output separately, and the final forecast is established by majority vote (*Cutler, Cutler & Stevens, 2012*).

- Logistic regression (LR) is a statistical ML technique used for binary classification tasks to predict the likelihood of an instance falling into a specific class. The logistic function (sigmoid function) describes the relationship between the characteristics and the binary outcome in logistic regression. Predictions based on LR yield discrete values best suited for binary categorization (*Boateng & Abaye, 2019*).

### The stacking model

The ensemble machine learning technique combines multiple models as base learners to provide predictions or judgments. It uses the diversity and accumulated knowledge of the base learners to improve the overall prediction performance (*Zounemat-Kermani et al., 2021*). Multiple models are combined in model stacking, and a meta-learner model is used to improve model predictions. The meta-learner minimizes the weaknesses of each model while maximizing its strengths. Using the k-fold cross-validation technique, a meta-model is built from the predictions of several ML base models (weak learners). Finally, the meta-model was trained using the "final estimator" or "final learner" ML model (*Cui et al., 2021*). As a result, overall performance can be improved and a model superior to any intermediate model can be obtained (*Sikora, 2015*).

The stacking model develops during two levels as shown in Fig. 4.

In level 1, heterogeneous weak learners learn in parallel using a training set. Then, the output of the training set is combined in stacking training, and the production of the testing set is combined in a stacking test.

In level 2, a meta-learner is trained using stacking training and evaluating using testing stacking to predict the final output.

### Explainable artificial intelligence

XAI concerns developing machine learning and artificial intelligence (AI) algorithms that justify their predictions or conclusions (*Barredo Arrieta et al., 2020*). The gap between the complicated inner workings of AI models and the desire for openness and interpretability in their outputs will be bridged with the help of XAI (*Ngo, Beard & Chandra, 2022*). One advantage of the XAI approach is understanding why an AI system made a particular decision, improving trust, accountability, and the ability to detect distortions or faults (*Tjoa & Guan, 2020*).

Ensemble models, in their nature, combine multiple algorithms with improving predictive performance, which makes them complex and opaque, making it challenging for stakeholders and users to understand how decisions are made (*Rane, Choudhary & Rane, 2024*). XAI plays a crucial role in addressing this issue by providing frameworks and techniques that reveal the interpretability of these models in a comprehensible manner by breaking down the contributions of individual models within the ensemble, highlighting how each one influences the final prediction (*Dwivedi et al., 2023*). XAI plays an essential role in improving the interpretability of ensemble models by analyzing information: XAI techniques can measure the contribution of each in the ensemble model's predictions by

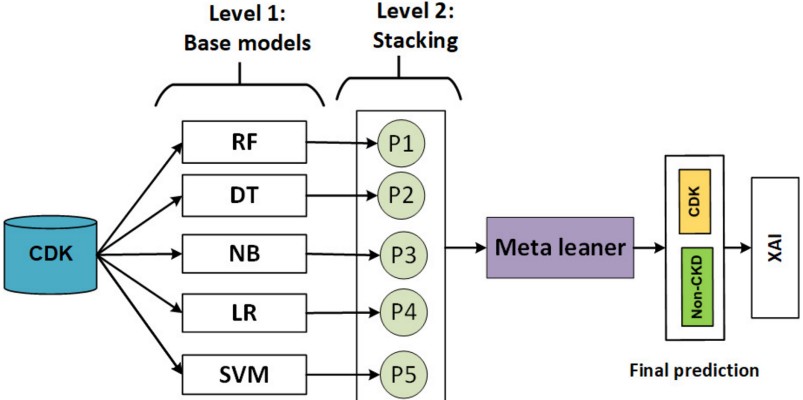

**Figure 4 Flowchart describing of the stacking model.**

identifying global feature importance and local importance. This helps users understand which of the most essential features play a vital role in decision-making in the ensemble. XAI tools can decompose the final output into the contributions of each base model in the ensemble, demonstrating how individual models (*Chamola et al., 2023*).

- Global exploitability aims to fully understand the behavior of an AI model in its entire input space (*Ding et al., 2022*). It focuses on discovering the model's biases, habits, or rules. Global exploitability approaches can be used to understand the structure of the model, its learned representations, and the relative importance of various aspects (*Ding et al., 2022*).

- Local explainability is concerned with providing reasons for specific predictions or decisions that an AI model makes. Attempts to discover the variables or features that contribute to a particular outcome. Local explainability techniques can illuminate the model's decision-making process by emphasizing the significance or contribution of various features for a single instance (*Ding et al., 2022*).

- SHAP (SHapley Additive exPlanations) is an XAI method that attempts to deliver information about machine learning models' predictions. It is based on the cooperative game theory notion of Shapley values. SHAP's fundamental principle is to assign a relevance score to each feature in a forecast. These scores represent how much each feature adds to the prediction compared to a reference baseline (*Marcílio & Eler, 2020*). Understanding the feature contributions provides insights into the model's decision-making process and allows for improved interpretation of its predictions (*Marcílio & Eler, 2020*).

### Evaluation models

Classification performance is typically measured using precision, recall, F1-score, and accuracy, Matthews correlation coefficient (MCC) and Cohen Kappa. True positive (TP), false positive (FP), true negative (TN), and false negative (FN) are calculated as true positives, false positives, and false negatives, respectively. While TN showed a negative

result, it actually returned a positive result, while TP showed a negative result, but it actually returned a positive result. TP indicates that the result is actually positive, whereas TN indicates that the result is actually negative.

$$Accuracy = \frac{TP + TN}{TP + FP + TN + FN}.$$  (1)

$$Precision = \frac{TP}{TP + FP}$$  (2)

$$Recall = \frac{TP}{TP + FN}$$  (3)

$$F1\text{-}score = \frac{2 \cdot precision \cdot recall}{precision + recall}$$  (4)

$$MCC = \frac{TP \cdot TN - FP \cdot FN}{\sqrt{(TP + FP)(TP + FN)(TN + FP)(TN + FN)}}$$  (5)

$$CohenKappa = \frac{(TP + TN) - E}{N - E}$$  (6)

where: $N = TP + TN + FP + FN$ and $E = \frac{(TP + FP)(TP + FN) + (FN + TN)(FP + TN)}{N}$

The receiver operating characteristic (ROC) is a graphical curve that evaluates the performance of binary classification at various thresholds. It shows the trade off between TPR and specificity.

## Online prediction pipelines

The real-time data pipeline is an intelligent and dynamic system that integrates intelligence methods and advanced technologies to store streaming data and extract information from high-volume data. It enables healthcare sectors to execute analytics and extract useful patterns from the streaming data generated continuously from different sources.

Ensuring the reproducibility and reliability of predictions through a real-time validation framework. Figure 1 presents the main steps of developing real-time analytics in stream processing frameworks designed to manage the continuous flow of data and make decisions. Firstly, data is generated from streaming resources and then ingested by Apache Kafka, efficiently capturing real-time data streams from various sources by providing fault tolerance and high throughput. Once the data is ingested into Kafka, it can be received and preprocessed immediately in real-time using big data streaming platforms. The models are loaded models for integrating ML models with big data streaming platforms, and the preprocessed data on ML models are applied to predict health status in real-time. Afterward, the model's results can be visualized on the dashboard or integrated into other reporting tools. This study introduces a real-time system for the early detection CKD, leveraging a real-world simulation environment built using Apache Kafka, Confluent Cloud, and Python scripts.

### Big data platforms

Online prediction pipelines were developed using Apache Kafka and the confluent platform.

- Kafka was used to build streaming apps and pipelines for real-time data. The Kafka system can receive large amounts of data in real-time and with low latency in addition to being real-time, fault-tolerant, and scalable. It (*Garg, 2013*) Streaming data can be read from sensors and ports. Kafka allows you to read and write streams of data. The development of real-time stream processing applications is possible. The data streams are stored in a distributed, replicated, and fault-tolerant cluster. There are two main libraries for APIs: Producer APIs and Consumer APIs. Kafka topics can be created using the Producer API. Kafka topics are subscribed to, and stream records are processed using the Consumer API.

- The Apache Kafka stream processing engine is an open-source, distributed system that offers the ability to filter, aggregate, and join data streams based on predefined patterns as an IaaS (Infrastructure-as-a-Service) service to generate predictive inferences (*Garg, 2013*). It supports publishing and subscribing to data streams, storing them fault-tolerantly, and processing them in real-time. Streaming engines process sensor data streams in real-time to predict event patterns based on sensor observations/readings and correlate the data with predefined/preset thresholds. As data streams from heterogeneous producers are processed *via* APIs in a fault-tolerant manner in a producer-publish, consumer-subscribe model, the platform is like an enterprise messaging system (*Garg, 2013*). The data streaming pipeline is processed and analyzed between Apache Kafka Connect APIs and heterogeneous systems or applications (*Cao et al., 2015*). KSQL is a streaming SQL engine to executes queries in real-time without pushing the data stream to the database. A data stream is a continuous flow of data records analyzed through the stream processing engine to generate an output stream (*Garg, 2013*). Sensors and IoT are utilized in smart cities, healthcare, and energy management to generate streaming data that is used to develop real-streaming predictive systems (*Cao et al., 2015*). A cluster of Apache Kafka servers can be installed locally or on the cloud, and each producer connects to the cluster through the Kafka Connector APIs. Topics are categories in which data records are stored. The records consist of a key, a value, and a timestamp. Input streams are converted into output streams using Kafka stream processing, which consumes input streams from available topics and produces output streams.

Kafka includes sets of features

- **Topic:** The Kafka cluster categorizes messages into topics based on their similar attributes. In an ordered list, topics are divided into partitions determined by an incremental identifier called an offset, which identifies messages/data in each partition. The Consumer reads the messages based on the chronological order or sequence of arrival generated by the Producer, where the messages are written to the appropriate topics (*Garg, 2015*).

- **Producers:** Apache Kafka producers are the sources of data and messages that write to topics, which are then written to partitions and stored on brokers. Sensors, devices, and manual or automatic systems produce events, which are a type of event producer (*Garg, 2015*).

- **Consumer:** A consumer subscribes to a topic to ingest or consume the messages/data generated by the producer (*Narkhede, Shapira & Palino, 2017*). Consumers are configured to read streaming data by automatically connecting to a suitable broker with read access. Consumers automatically fetch data from the next available broker when there is a failure of one of the brokers in the cluster. According to the offsets, data/messages are consumed in order. Streaming data offsets provide consumers with information about where to start reading a topic's streaming data. Data/messages are read from multiple partitions of the same topic in parallel when there is more than one partition (*Narkhede, Shapira & Palino, 2017*).

- The confluent platform is used to develop fully streaming pipelines based on Apache Kafka (*Confluent, 2014*). It provides real-time streaming of data on a scalable platform, making the information flow seamless. Apache Kafka is integrated with it, allowing efficient data processing, storage, and streaming. In addition, it provides scalable solutions to accommodate growing volumes of data, making it possible for the platform to handle more workloads and a greater number of demands over time (*Confluent, 2014*). A fault-tolerant mechanism is integrated into the Confluent platform, improving workflow reliability and resilience. The streaming platform provides connectors and APIs for integrating data sources and systems. Platform administrators can effectively monitor and manage the streaming infrastructure with management and monitoring tools (*Confluent, 2014*).

### Online prediction pipelines steps

Numerous data sources in the healthcare industry continuously produce real-time data from linked equipment, including medical devices: Wearable sensors, such as glucose and heart rate sensors, and smart homes, which are IoT smart appliances, security cameras, and thermostats. Kafka is used to consume and produce these data. Different types of pre-processing steps will be applied, such as filling in missing values, converting data formats to maintain uniformity and normalize numerical values. In our work, online prediction pipelines consist of the following steps:

The Confluent Cloud and Apache Kafka service are integrated with a Python application to develop online prediction pipelines. Python applications can produce and consume messages using a Confluent Cloud and Apache Kafka cluster. The process involves creating an account on Confluent Cloud, setting up a Kafka cluster, and creating Kafka topics. API keys are then generated for authentication purposes. These keys enable Python applications to securely connect to the Confluent Cloud and Apache Kafka clusters.

- Set up Confluent Platform: We create an account on the Confluent Platform, then create a cluster and topics, and generate keys. These keys are used to connect to topics, allowing a Python script to produce and consume messages using the keys.
- A Python script is being developed to generate streaming health attributes. A vast range of health attribute data is generated and pushed into a Kafka topic in using the Producer API (confluent_kafka.Producer). The script buffers incoming data streams from the producers and transfers the data to the topic. This results in improved fault tolerance due to load balancing in the event of component failures.
- A Python script has been developed to consume health attributes from a Kafka topic using the Consumer API (confluent_kafka.Consumer). The consumed health attributes are then applied different pre-processing steps Impute missing values mean, median, mode and then converted to the same format as the input to the model. Subsequently, the model is loaded and applied to the health attributes for real-time prediction. Finally, the prediction result is saved to a topic.

## RESULTS

### Experimental setup

Python was used to conduct the offline model phase. The analysis relied on the following libraries and coding packages: (1) Scikit-Learn (*Pedregosa et al., 2011*), (2) Matplotlib (*Hunter, 2007*), and (3) the sklearn-genetic library is used to implement genetic feature selection methods (*Calzolari, 2019*). (4) The PySwarms library implements swarm feature selection methods. A summary of the PSO and GA parameters for selecting the best features can be found in Table 3. Table 4 shows the parameter settings for each ML model. Encoding type is binary vector representation (1 = feature selected, 0 = feature not selected). The fitness function uses accuracy as the evaluation criterion for selected feature subsets. The decision variables of PSO problem are constrained within the interval [0, 1], where each variable represents the probability of selecting a feature, and only features with values greater than 0.1.

The Confluent Cloud Kafka service is integrated with a Python application to develop online prediction pipelines. Confluent Cloud is used to create accounts, Kafka clusters, and topics to generate key and PI keys for authentication. Python applications will use these keys to connect securely to the Kafka cluster and topic.

### Results of offline phase

This section presents the results of models employing GA and PSO in various experiments based on different dataset splits, including Experimental 1 (80% training set and 20% testing set), Experimental 2 (70% training set and 30% testing set), and Experimental 3 (60% training set and 40% testing set).

#### Experimental 1

Table 5 shows a comprehensive evaluation of ML models (RF, LR, DT, SVM, and NB) and stacking models based on two different types of FS methods, namely PSO and GA, on the task of predicting CKD with a set of tests 20% using evaluation metrics: accuracy,

**Table 3 Parameters setting of FS methods for GA and PSO.**

| FS methods | Parameters | Values |
|---|---|---|
| GA | Crossover rate | 0.5 |
| | Mutation rate | 0.2 |
| | Population size | 100 |
| | Iteration number | 100 |
| | Population | 100 |
| PSO | Population size | 20 |
| | Max number of generation | 30 |
| | Early stopping | True |
| | Local best weight | 1 |
| | Global best weight | 1 |
| | Use local random seed | True |

**Table 4 Parameter settings for each of ML models.**

| Models | Parameters | Values |
|---|---|---|
| RF | n_estimators | 100 |
| | max_depth | 3 |
| | min_samples_split | 2 |
| | min_samples_leaf | 1 |
| | bootstrap | True |
| LR | penalty | l2 |
| | C | 1.0 |
| | max_iter | 100 |
| NB | var_smoothing | 1e−9 |
| SVM | C | 1.0 |
| | kernel | rbf |
| DT | max_depth | 4 |
| | min_samples_split | 2 |
| | min_samples_leaf | 1 |
| | criterion | Gini |

precision, recall and F1-score and AUC and two statistical analysis Kappa and MCC. Based on the GA, the stacking model achieves 100 as the best results compared to other models and scores 100 on both Kappa and MCC because it enhances results by integrating the output of base models with meta-learner to strengthen performance and generalization. NB records the worst performance with 92.50 accuracy, 92.60 F1-score, and 84.615 Cohen Kappa for GA, 93.75 accuracy, 93.83 F1-score and 87.09 Cohen Kap for PSO. The second-best performance is conducted by RF and SVM for GA with accuracy, precision, recall, and F1-score at 98.75, 98.79, 98.75, and 98.75, respectively, because RF combines multiple decision trees to reduce overfitting and enhance performance. Based on GA, RF, SVM, scores very high scores (97.3–97.4) on both Cohen and Kappa MCC.

**Table 5 The results of the models using two feature selection methods: GA and PSO are based on a 20% testing set.**

| Feature selection methods | Models | Accuracy | Precision | Recall | F1-score | Cohen Kappa | MCC |
|---|---|---|---|---|---|---|---|
| GA | RF | 98.75 | 98.79 | 98.75 | 98.75 | 97.35 | 97.38 |
| | LR | 97.50 | 97.66 | 97.50 | 97.51 | 94.73 | 94.86 |
| | DT | 97.50 | 97.66 | 97.50 | 97.51 | 94.73 | 94.86 |
| | SVM | 98.75 | 98.79 | 98.75 | 98.75 | 97.35 | 97.38 |
| | NB | 92.50 | 92.75 | 92.50 | 92.60 | 84.61 | 85.63 |
| | Stacking | 100.00 | 100.00 | 100.00 | 100.00 | 100 | 100 |
| POS | RF | 97.50 | 97.66 | 97.50 | 97.51 | 94.73 | 94.86 |
| | LR | 97.50 | 97.66 | 97.50 | 97.51 | 94.73 | 94.86 |
| | DT | 96.25 | 96.31 | 96.25 | 96.26 | 92.05 | 92.08 |
| | SVM | 97.50 | 97.66 | 97.50 | 97.51 | 94.73 | 94.86 |
| | NB | 93.75 | 94.64 | 93.75 | 93.83 | 87.09 | 87.83 |
| | Stacking | 98.75 | 98.79 | 98.75 | 98.75 | 97.35 | 97.38 |

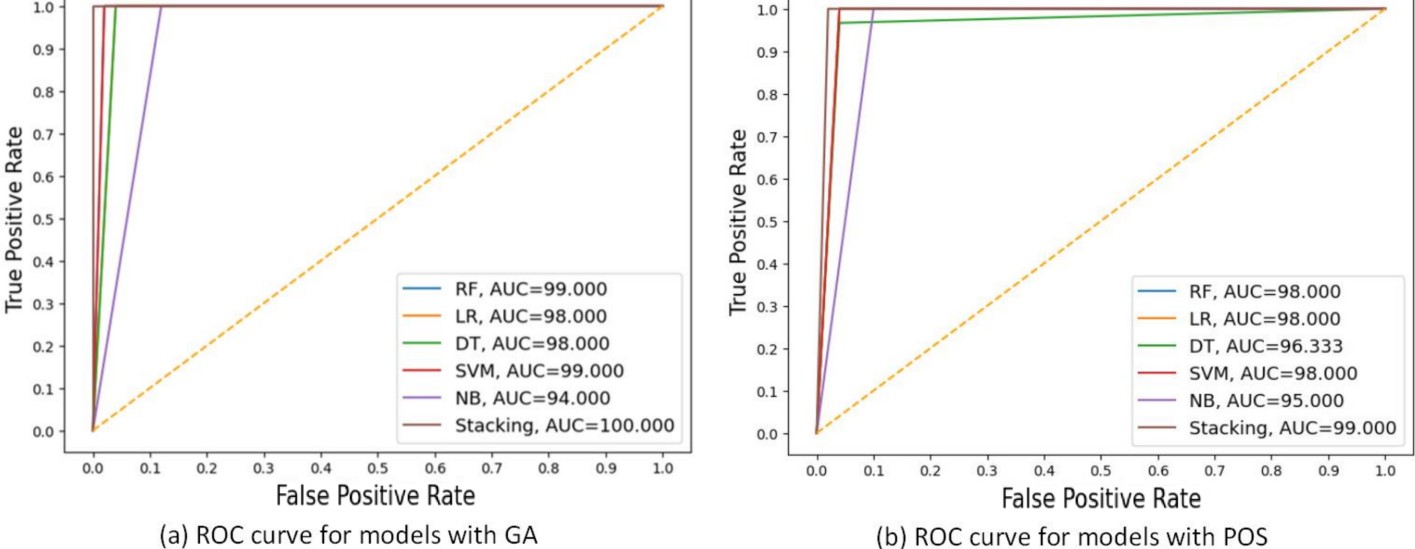

(a) ROC curve for models with GA

(b) ROC curve for models with POS

**Figure 5 ROC curves comparing the performance of different models using two feature selection techniques: (A) GA and (B) PSO.** The models include RF, LR, DT, SVM, NB and stacking ensemble based 20% of testing set. AUC values are shown for each model, indicating that the Stacking model got the highest AUC in both scenarios, with the best AUC = 100.000 using GA.

Figure 5 shows the ROC curves for these algorithms based on the area under the curve (AUC) metric. As the AUC value increases, the model's generalization ability increases, as indicated on the ROC curve by the curve close to the upper-left corner. NB performed the worst, with an AUC value of 94 for GA and 95 for PSO. The stacking model had the highest AUC value of 100 for GA.

**Experimental 2**

Table 6 compares the effect of using two different FS methods, GA and PSO, based on various models: stacking, RF, LR, DT, SVM, and NB. with a 30% testing set. The

**Table 6 The results of the models using two feature selection methods: GA and PSO are based on a 30% testing set.**

| Feature selection methods | Models | Accuracy | Precision | Recall | F1-score | Cohen Kappa | MCC |
|---|---|---|---|---|---|---|---|
| GA | RF | 97.67 | 97.94 | 97.67 | 97.69 | 94.73 | 94.86 |
| | LR | 95.95 | 95.95 | 95.95 | 95.95 | 91.22 | 91.35 |
| | DT | 95.45 | 95.55 | 95.45 | 95.55 | 91.15 | 91.16 |
| | SVM | 97.02 | 97.02 | 97.02 | 96.02 | 93.01 | 93.24 |
| | NB | 95.83 | 95.09 | 95.83 | 95.78 | 91.30 | 91.65 |
| | Stacking | 99.17 | 99.18 | 99.17 | 99.17 | 98.23 | 98.24 |
| POS | RF | 97.11 | 97.15 | 97.11 | 97.15 | 92.82 | 92.88 |
| | LR | 94.43 | 94.25 | 94.43 | 94.47 | 89.42 | 89.48 |
| | DT | 94.17 | 94.36 | 94.17 | 94.20 | 87.71 | 87.84 |
| | SVM | 96.67 | 96.75 | 96.67 | 96.68 | 93.01 | 93.24 |
| | NB | 93.44 | 93.01 | 93.44 | 93.01 | 84.34 | 84.66 |
| | Stacking | 97.80 | 97.70 | 97.80 | 97.70 | 94.64 | 94.65 |

comparison relies on measuring each model's achieved accuracy, precision, recall, and F1-Score, and two statistical analysis Kappa and MCC.

For GA, the stacking model achieves the best performance compared to other models, with accuracy at 99.17%, precision at 99.18, recall at 99.17, and F1-score at 99.17, Cohen Kappa at 98.23, and MCC at 98.24. DT and LR record approximately the same percentage of each score at 95%. DT records lowest at 95.83, 95.09, 95.83, and 95.78 of accuracy, precision, recall, and F1-score, respectively. For PSO, the stacking model records the best performance compared to ML models with 97.80 accuracy. DT, LR and NB record approximately the same percentage at 94%. NB with PSO records the worst performance compared to other models at 93.44, 93.01, 93.44, and 93.01, 84.34, and 84.66 in accuracy, precision, recall, F1-score, Cohen Kappa and MCC respectively.

Figure 6A displays the ROC curve, and AUC values of each model with GA and PSO. It can be observed that stacking achieves the highest AUC at 99.33 for GA, indicating the best discriminatory ability in distinguishing between the positive and negative classes. While NB had the lowest AUC at 91 for PSO.

### Experimental 3

Table 7 shows a comprehensive evaluation of ML models (RF, LR, DT, SVM, and NB), stacking models based on two different types of FS methods, namely PSO and GA, on the task of predicting CKD with a set of tests 40% using evaluation metrics and two statistical analysis: Cohen Kappa and MCC. Based on the GA, the stacking model achieved 98.75, 98.77, 98.75, 98.75, 97.333 and 97.333 as the best results of accuracy, precision, recall, F1-score, Cohen Kappa, and MCC compared to other models because it enhances results by integrating the base models with meta-learner to strengthen performance and generalization. The second-best performance is conducted by RF for GA with accuracy, precision, recall, F1-score, Cohen Kappa, and MCC at 96.88, 96.89, 96.88, 96.88, 93.355, and 93.364, respectively, because RF combines multiple decision trees to reduce overfitting and enhance performance. DT records the worst performance with 91.88 accuracy 91.95
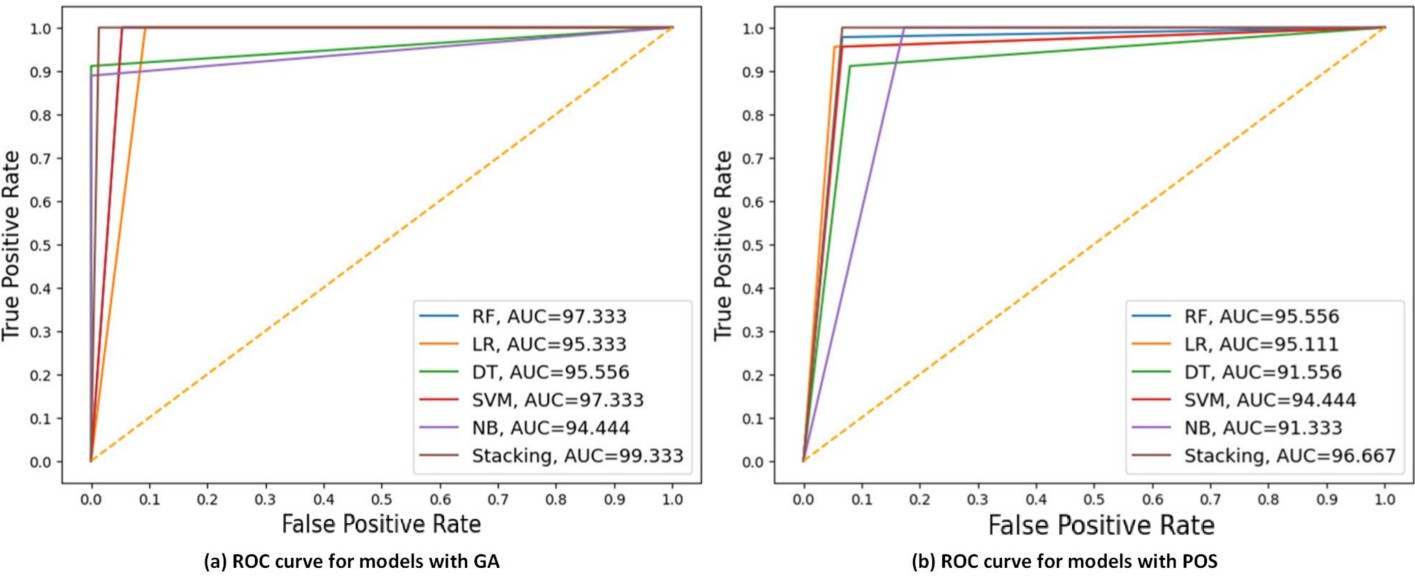

(a) ROC curve for models with GA

(b) ROC curve for models with POS

**Figure 6 ROC curves compare the performance of different models using two feature selection techniques: (A) GA and (B) PSO.** The models include RF, LR, DT, SVM, NB, and stacking model based on 30% of the testing set. AUC values are shown for each model, indicating that the Stacking model got the highest AUC in both scenarios, with the best AUC = 99.33 using GA.

**Table 7 The results of the models using two feature selection methods: GA and PSO are based on a 40% testing set.**

| Feature selection methods | Models | Accuracy | Precision | Recall | F1-score | Cohen Kappa | MCC |
|---|---|---|---|---|---|---|---|
| GA | RF | 96.88 | 96.89 | 96.88 | 96.88 | 93.35 | 93.36 |
| | LR | 95.62 | 95.74 | 95.62 | 95.65 | 90.75 | 90.83 |
| | DT | 91.88 | 92.39 | 91.88 | 91.95 | 81.81 | 82.26 |
| | SVM | 96.25 | 96.42 | 96.25 | 96.27 | 92.05 | 92.08 |
| | NB | 94.38 | 95.11 | 94.38 | 94.44 | 87.09 | 87.83 |
| | Stacking | 98.75 | 98.77 | 98.75 | 98.75 | 97.33 | 97.33 |
| POS | RF | 95.62 | 95.65 | 95.62 | 95.63 | 89.54 | 89.81 |
| | LR | 94.38 | 94.40 | 94.38 | 94.38 | 87.96 | 87.96 |
| | DT | 90.50 | 90.62 | 90.50 | 90.70 | 80.06 | 80.07 |
| | SVM | 93.75 | 93.95 | 93.75 | 93.79 | 87.96 | 87.96 |
| | NB | 91.88 | 92.39 | 91.88 | 91.95 | 84.10 | 84.13 |
| | Stacking | 96.25 | 96.31 | 96.25 | 96.26 | 92.053 | 92.085 |

F1-score, and 81.818 Cohen Kappa for GA. Based on PSO, the stacking model records the best performance with 96.25 accuracy, 96.26 F1-score, and 92.053 MCC. DT records the worst with 90.50 of accuracy, 90.70 of F1-score, and 80.07 of MCC.

Figure 7 displays the ROC curve and AUC values of each model with GA and PSO. It can be observed that stacking achieves the highest AUC at 98.667 for GA, indicating the best discriminatory ability in distinguishing between the positive and negative classes. DT had the lowest AUC at 90 for PSO.

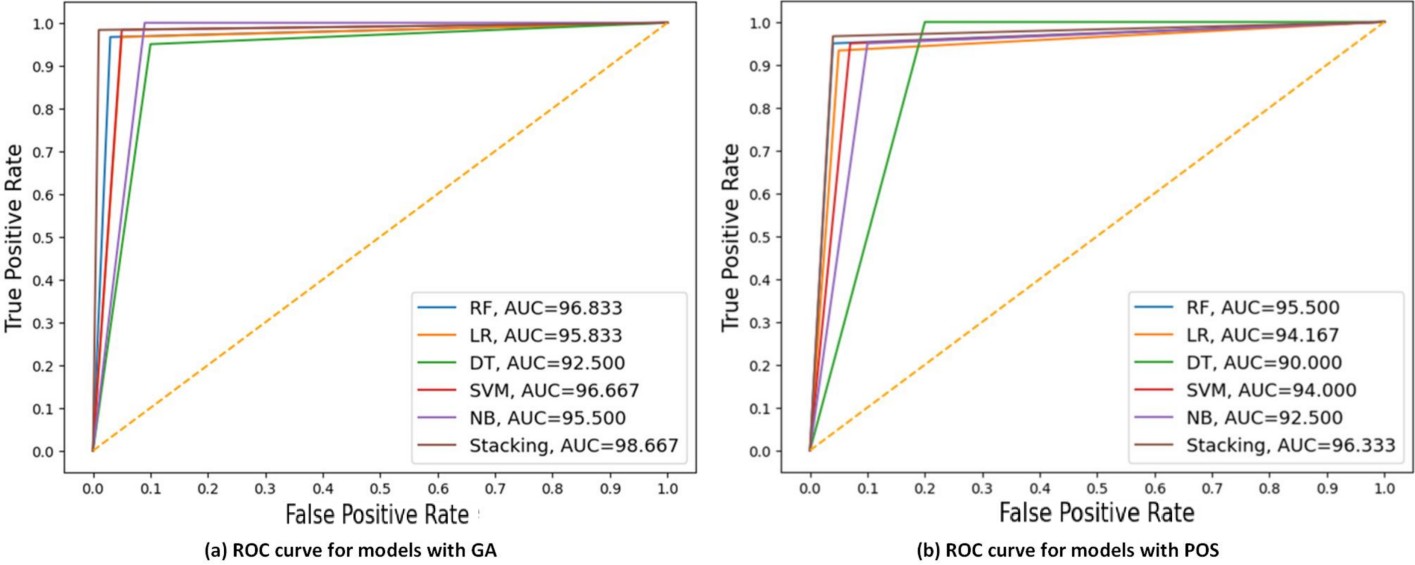

**Figure 7** **ROC curves compare the performance of different models using two feature selection techniques: (A) GA and (B) PSO.** The models include RF, LR, DT, SVM, NB, and stacking model based on 40% of the testing set. AUC values of models are presented, indicating that the stacking model got the highest AUC in both scenarios, with the best AUC = 98.667 using GA.

## The steps of integrating model with big data platforms

This section provides the main steps of integrating the stacking model with big data streaming platforms (Confluent Cloud and Apache Kafka) to develop a real-time application system. The results from the offline phase showed that the stacking model with GA achieved the highest performance that is used to predict CKD in real-time. A fully streaming processing pipeline has been developed based on different APIs that support Confluent Cloud using Apache Kafka. The steps are outlined as follows:

(1) Creating an account on Confluent Cloud to create a cluster called 'Cluster_0' as shown in Fig. 8.

(2) Creating a topic called 'topic_0' as shown in Fig. 9 to generate keys used to connect cluster and topic.

(3) Python script is developed to generate streaming health attributes (specific gravity, albumin, serum creatine, hemoglobin, hypertension, coronary artery diseases, appetite, and anemia) that push into a Kafka topic as JSON format using the Producer API (confluent_kafka.Producer). The script buffers incoming data streams from the producers and transfers the data to the topic, as illustrated in Algorithm 1. An example of health attributes is pushed into 'topic_0,' as depicted in Fig. 10. Figure 11 shows the number of messages pushed to topic_0 and read from topic_0.

(4) A Python script is developed to consume health attributes from a Kafka topic using the Consumer API (confluent_kafka.Consumer). The consumed health attributes are then converted to the same format as the input to the model. Subsequently, the stack model is loaded and applied to the health attributes for real-time prediction, and the result of
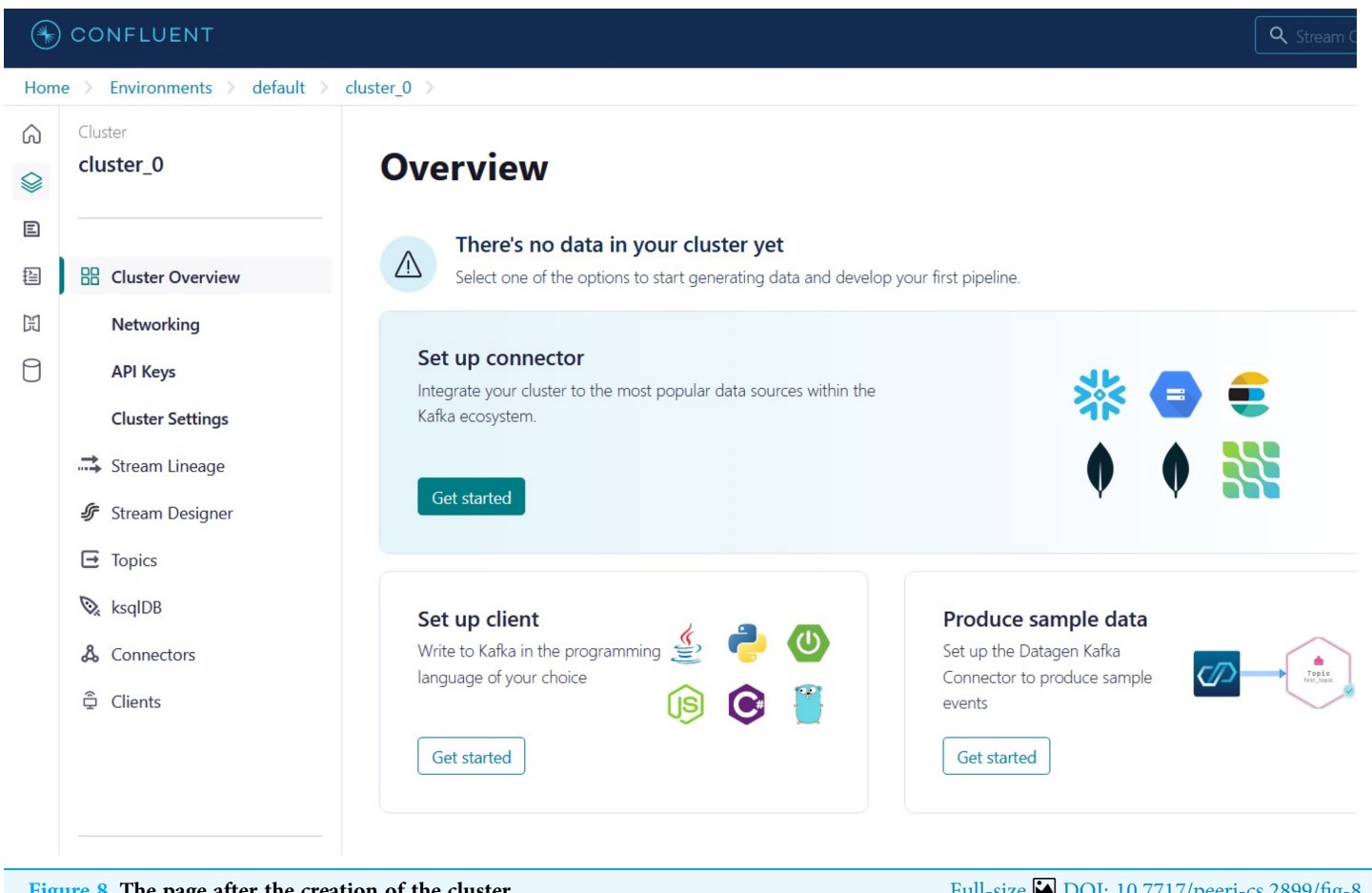

**Figure 8 The page after the creation of the cluster.**               

the predicted value is pushed in topic_1 as illustrated in Algorithm 2. Figure 12 shows examples of the predicted values pushed to topic_1.

## DISCUSSION

Different experimenters using different dataset splits are conducted, including Experimental 1 (80% training set and 20% testing set), Experimental 2 (70% training set and 30% testing set), and Experimental 3 (60% training set and 40% testing set). In conclusion, according to all our experiments, we noticed an increasing performance in terms of accuracy metrics when relying stacking. According to the results, the accuracy of the best models has increased from 98.75 in the regular ML to 100 in the stacking models. These results proved that the stacking model models outperformed all other ML models. Figure 13 compares models with GA and PSO using 20%, 30%, and 40% of testing set that shows stacking model model with GA records the best performance at 100, 99.17, and 98.75, respectively. "Model interpretation using XAI" presents the model interpretation using XAI for stacking the model with GA.

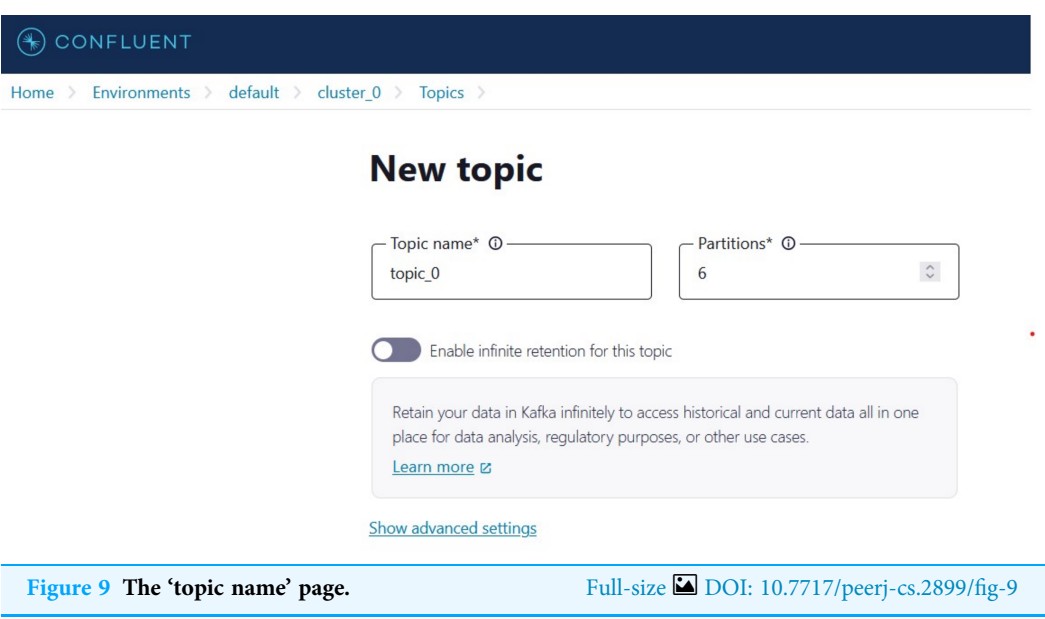

**Figure 9** The 'topic name' page.               

---

| **Algorithm 1**   Pseudocode of consumer. |
| --- |

Step 1: Loading model

Step 2: Define a set of configurations, including the server name and keys,

for accessing Kafka Producer and Kafka Consumer instances.

Step 3: Define consumer to read a message from the topic

consumer = Consumer (config)

consumer.subscribe (['topic_0'])

Step 4: Define Kafka Producer to push prediction results in topic_1

def predict_disease (data):

      model = joblib.load ('model.joblib')

try:

      while True:

      Reading a message from topic_0.

      Extracting health attributes.

      Converting health attributes into the same format for the model.

      Applying health attributes to the model to predict a value.

      Pushing the predicted value to topic_1.

except KeyboardInterrupt:

      pass

finally:

      consumer.close()

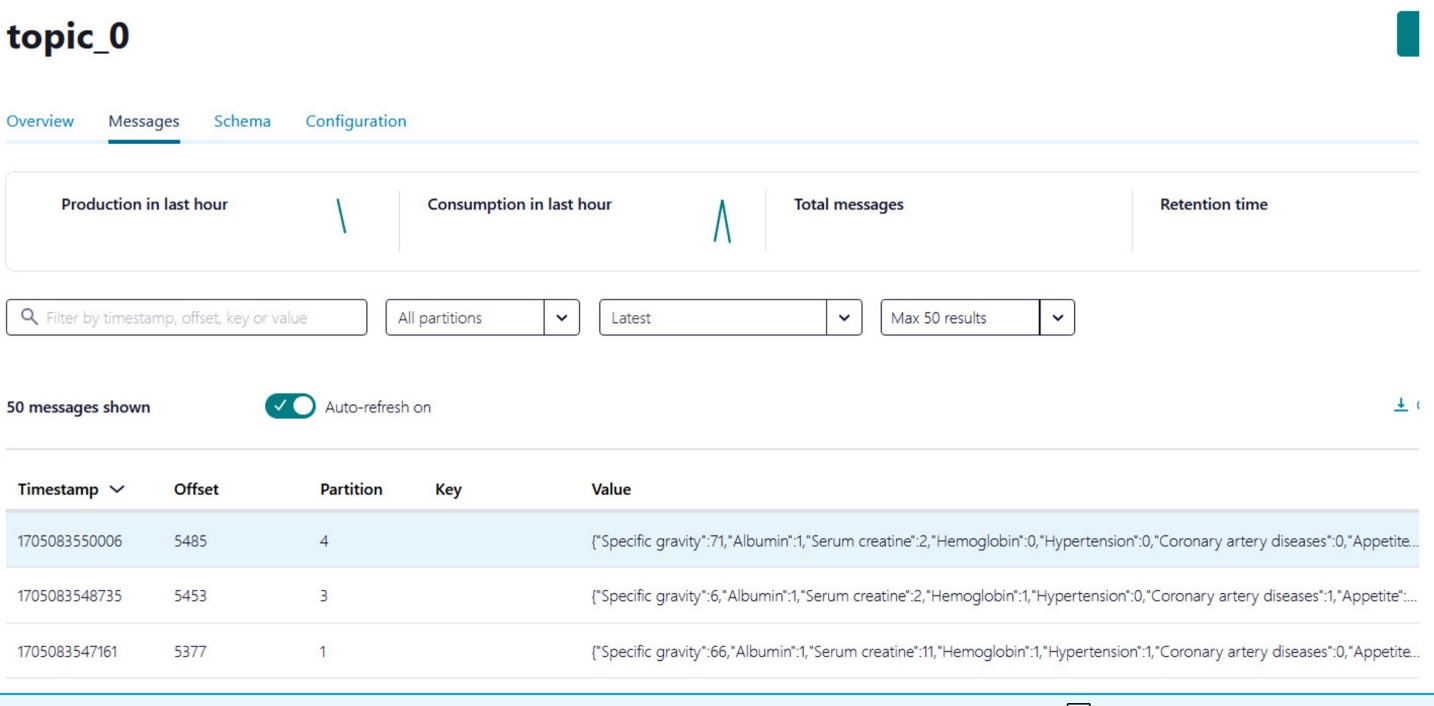

**Figure 10** The example of health attributes is pushed in topic_0.

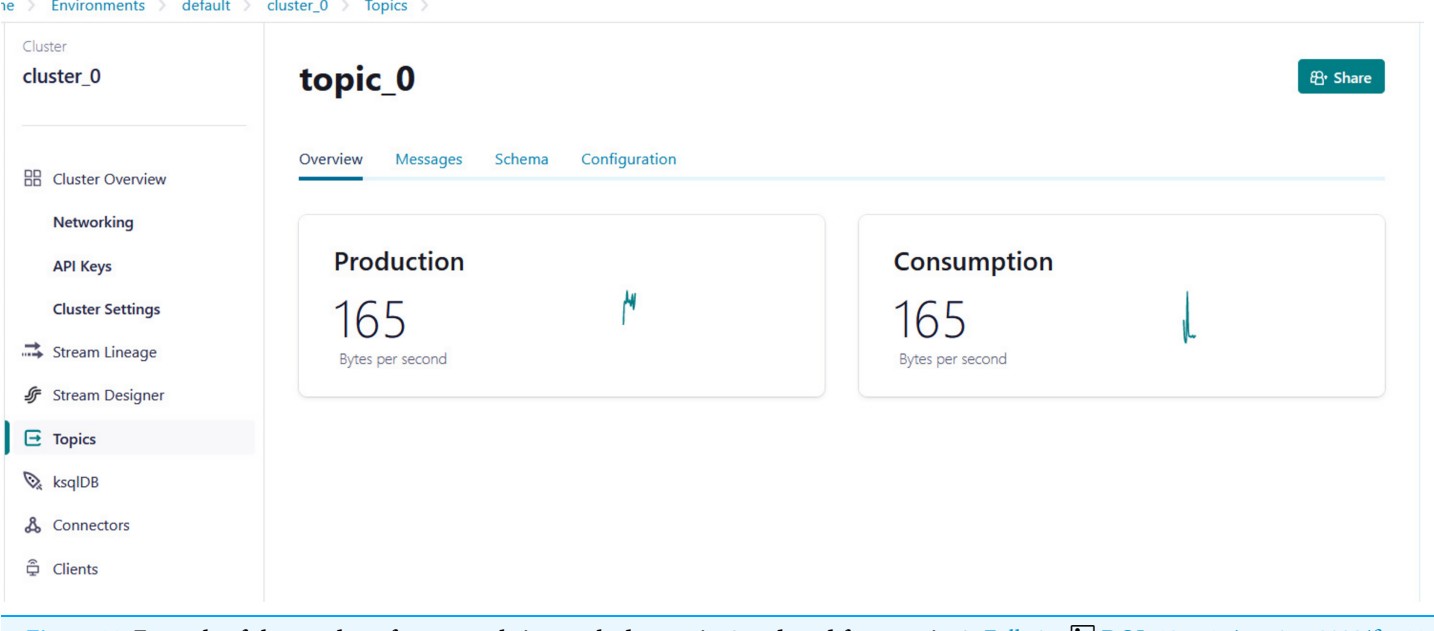

**Figure 11** Example of the number of messages being pushed to topic_0 and read from topic_0.

| Algorithm 2 | Pseudo code of producer. |
|---|---|

Step 1: Define a set of configurations, including the server

name and keys, for accessing Kafka Producer

Step 2: Define Kafka Topic

producer = Producer (config)

topic = 'topic_0'

Step 3: Define a function to generate health attributes

def health_attributes():

    return {

        'Specific gravity': random.randint (1, 100),

        'Albumin': random.choice ([0.0, 1.0]),

        'Serum creatine': random.randint (1, 15),

        'Hemoglobin': random.choice ([0.0, 1.0]),

        'Hypertension': random.choice ([0.0, 1.0]),

        'Coronary artery diseases': random.choice ([0.0, 1.0]),

        'Appetite': random.choice ([0.0, 1.0]),

        'Anemia': random.choice ([0.0, 1.0]),

    }

try:

    while True:

        Generate health_attributes

        Produce a message on the topic

        Flush the producer to ensure the message is sent

        Wait for a random interval before sending the next message

except KeyboardInterrupt:

    pass

finally:

    Close the producer

## Statistical testing to validate model superiority

We compared statistical different between models based on McNemar's Test (*Lachenbruch, 2014*) and Wilcoxon signed rank test (*Demšar, 2006*). McNemar's test is a statistical test used to compare the performance of two classification models on the same dataset based on compare prediction values between two models. Based on McNemar's test, As expected, stacking ensemble achieved a statistically significant difference compared with most of the models with $p$-value < 0.05 such as LR (0.031250), DT (0.004181), and NB (0.038574). However, as expected, the stacking achieved comparable performance with RF as it is an ensemble model of decision trees. Surprisingly, the stacking model did not achieve better results than SVM. There are many justifications for this, and one of them is

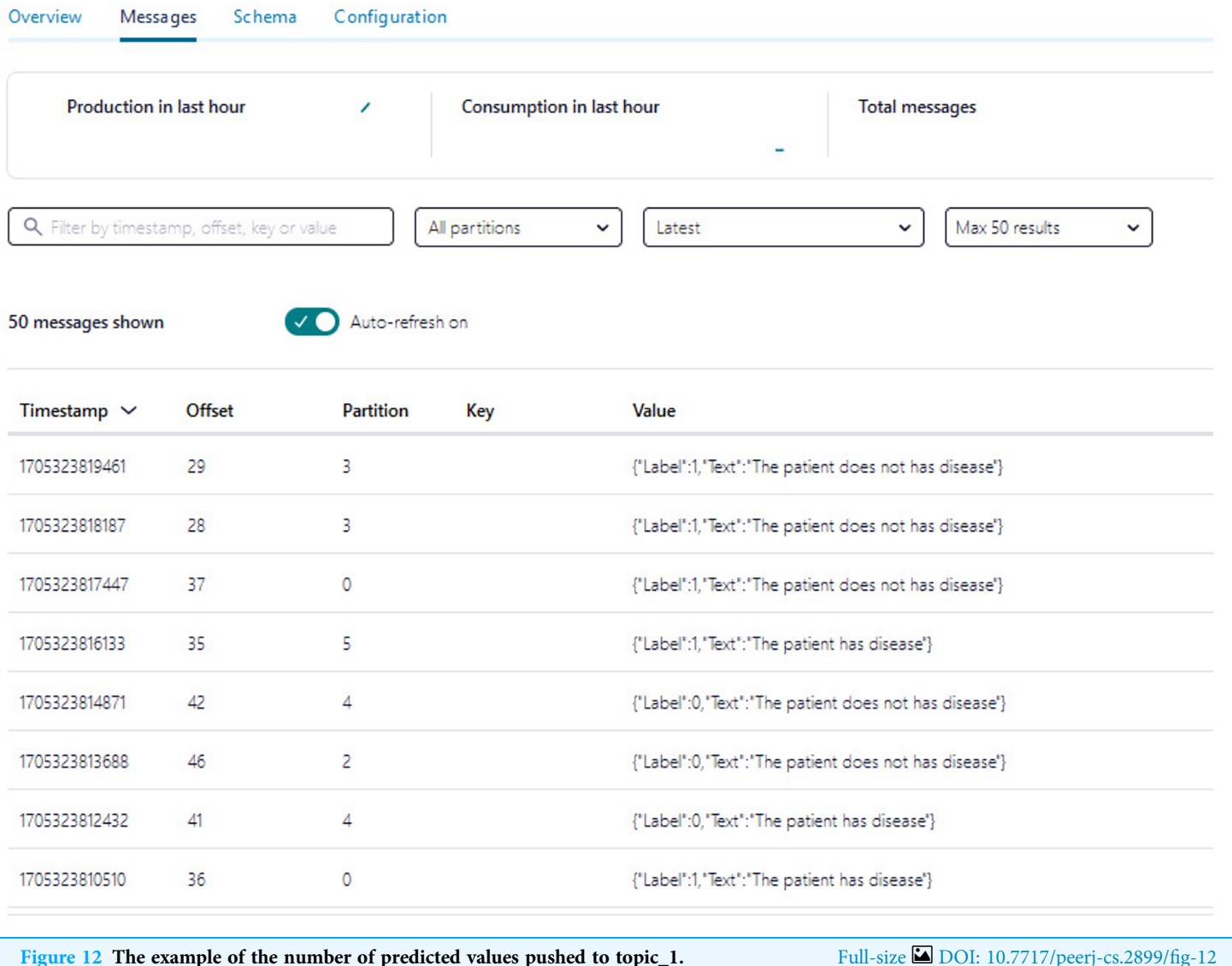

**Figure 12** The example of the number of predicted values pushed to topic_1.

because SVM is a strong classifier that learns the largest margin between the classes. However, the stacking model achieved statistically better results than most other models.

To establish the superiority of the proposed model, we employed the Wilcoxon signed rank test (*Demšar, 2006*), which is a widely used nonparametric statistical test for comparing the performance of algorithms. That test is utilized to specify the number of ties and wins between the tested models. If a model achieves a greater number of wins than ties, it is considered superior to the others.

In order to order and rank all classifiers and identify the best-performing one, we utilized the Nemenyi test (*Brown & Mues, 2012*). The average rank was calculated for each

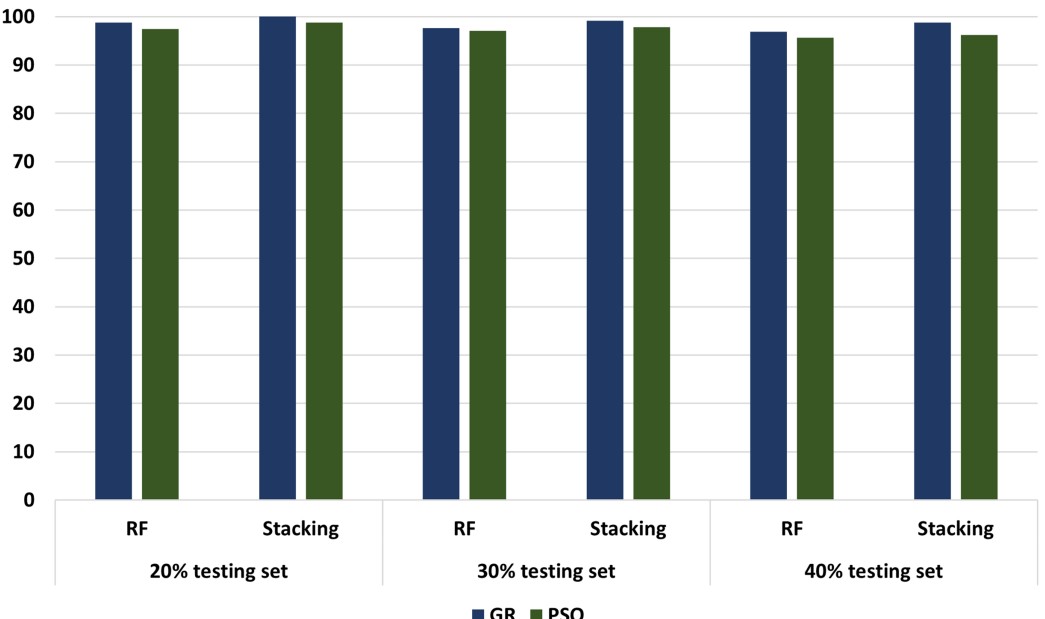

**Figure 13 The comparison of RF and stacking model with PSO and GA using 20%, 30% and 40% of testing set.**

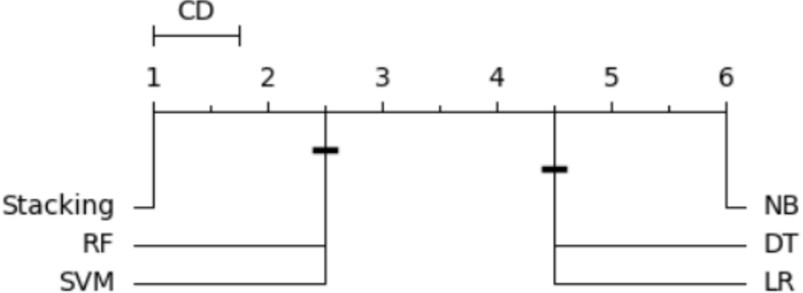

**Figure 14 Statistical tests of models using a 20% testing set.**

classifier based on its accuracy. When multiple classifiers are compared using the same dataset, the Nemenyi test results can be visualized based on critical distance. Figure 14 is a critical difference (CD) diagram that presents the average rank and critical difference between the traditional ML models and the stacking model from 1 to 6, ranking 1 as the best model and 6 as the worst model. This finding supports the notion that the stacking outperforms the existing approaches regarding accuracy and effectiveness.

## Model interpretation using XAI

Model explanations are crucial in understanding models, especially in healthcare; medical experts may have reservations about relying entirely on them. It would be helpful for healthcare professionals if they were able to identify the features that affect patient risk so that they could provide in-time and accurate decisions. Predictions provided insight into

feature-level features. To establish trustworthiness, several vital aspects must be understood, including the reasoning behind decisions and the influence of features. It is necessary to thoroughly evaluate the medical adequacy of these features. In the following section, we tried to explain the developed decision comprehensively by shedding light on our chosen ensemble classifier, which exhibits the highest accuracy. To provide model interpretation, we employ the SHAP (Shapley Additive explanations) is an explainable XAI method that attempts to deliver information about machine learning models' predictions. It is based on the cooperative game theory notion of Shapley values. SHAP's fundamental principle is to assign a relevance score to each feature in a forecast. These scores represent how much each feature adds to the prediction compared to a reference baseline (*Marcílio & Eler, 2020*). Understanding the feature contributions provides insights into the model's decision-making process and allows for improved interpretation of its predictions (*Marcílio & Eler, 2020*).

• Global exploitability: Figure 15 shows the summary plot of the combination feature important with feature effects. The features are presented on the y-axis, and the Shapley value of each instance is presented on the x-axis. A feature's value is represented by the color from low to high. Shapley values per feature are distributed along the y-axis by overlapping points. Features are ranked based on their importance, from high to low. Hemoglobin feature has the highest Shapley value, the most important feature. The second important feature is sg. Cad and ane features have the lowest Shapley values. In the summary plot, the impact of a feature on prediction is shown with its value.

### Local exploitability

The waterfall plots in SHAP provide descriptive information about individual predictions. In SHAP waterfall plots, red bars indicate positive contributions, while blue bars indicate negative contributions. On the right, you can see the importance of the feature, and on the left, you can see the value of the feature. Figure 16 illustrates the prediction of the model as $f(x)$, and the positive bias or intercept is displayed below the plot as $E[f(x)]$. Then, for each prediction, a red or blue line shows how the positive (red) or negative (blue) contribution of the feature affects the model output.

In Fig. 16A, the length of the corresponding bar represents the importance of each feature in the observation. Hemoglobin has the highest contribution with a SHAP value of +2.49. Specific gravity has the second-highest contribution with a SHAP value of +2.22. Serum creatine has the third-highest contribution with a SHAP value of +1.69.

In Fig. 16B, the red bars show the features that affect positively predicted values. Hemoglobin has the highest positive contribution with a SHAP value of +1.92. Hemoglobin has the second-highest positive contribution with a SHAP value of +1.38. The blue bars show the features that affect negatively predicted values. Albumin has the highest negative contribution with a SHAP value of −2.23. Hypertension has the second-highest positive contribution, with a SHAP value of −1.09.
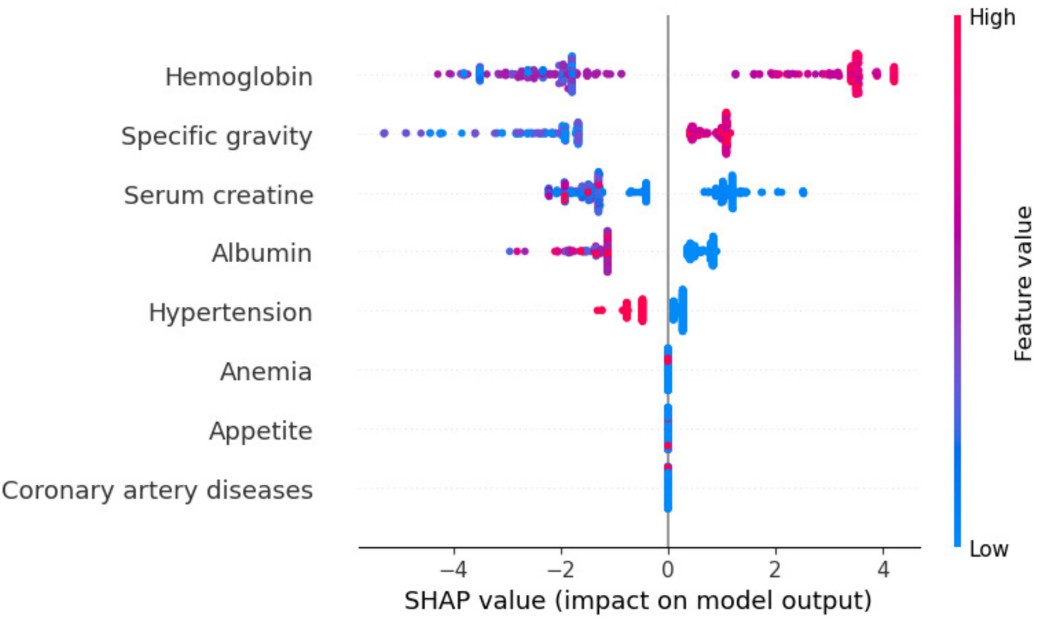

**Figure 15 Summary plot of the combination feature important with feature effects.**

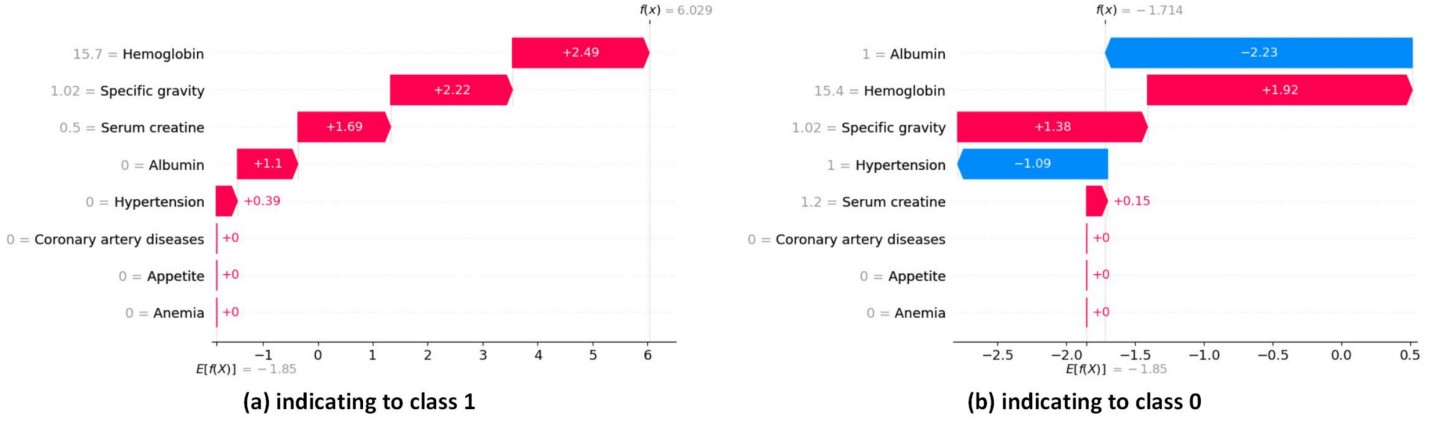

**Figure 16 SHAP local explanation according to each instance indicating to class 0 and class 1.**

## Real-world applications of integrated ML models in healthcare

– Early warning systems: The models can be used in hospitals to predict critical events such as sepsis, cardiac arrest, or rapid disease progression, allowing for effective medical intervention.
– Integration with electronic health records (EHRs): Models can be integrated with EHR system to predict data in real-time for providing clinical decisions.
– Mobile health applications: The model can be employed in mobile apps to allow patients tracking their health problem, and manage chronic diseases more effectively.

**Table 8 Comparison with existing work and the proposed models based on models, performance, and streaming platforms.**

| Articles | Models | FS | Accuracy | precision | Recall | F1-score | Streaming platforms |
|---|---|---|---|---|---|---|---|
| *Chittora et al. (2021)* | SVM | Wrapper | 98.86 | 98.59 | 97.22 | 97.90 | No |
| *Qin et al. (2019)* | Hybrid model | – | 99.83 | 99.84 | 99.80 | 99.86 | No |
| *Polat, Danaei Mehr & Cetin (2017)* | SVM | Filter | 98.5 | 98.6 | 98.5 | 98.88 | No |
| *Almasoud & Ward (2019)* | XGBoost | Filter | 99.10 | 99.5 | 99.33 | 99.10 | No |
| *Dritsas & Trigka (2022)* | RF | – | 99.2 | 99.20 | 99.20 | 99.20 | No |
| *Krishnamurthy et al. (2021)* | RF | – | 99.75 | 99.75 | 99.75 | 99.75 | No |
| Our work | Stacking model | GA | 100 | 100 | 100 | 100 | Confluent Cloud using Apache Kafka |

– Continuous learning systems: The models can be configured to continuously learn from current patient information, improving their accuracy through period of time and adapting to changes in medical treatments or overall wellness trends.

– Home-based health monitoring: The models can be included into internet of things (IoT) or smart home systems for tracking older or chronically ill patients, reducing hospital stays and enabling early diagnosis of health decline.

## Comparison with literature studies

This section compares our work with existing studies, as summarized in Table 8. The comparison is based on feature selection methods, models, accuracy, precision, recall, F1-score, and using streaming processing platforms. Table 8 shows that the existing studies did not address streaming processing platforms based on Confluent Cloud using Apache Kafka to predict CKD in real-time (*Chittora et al., 2021*; *Qin et al., 2019*; *Gunarathne, Perera & Kahandawaarachchi, 2017*; *Polat, Danaei Mehr & Cetin, 2017*; *Almasoud & Ward, 2019*). In contrast, our work introduces a stacking model with a GA and develops a real-time system using Confluent Cloud with Apache Kafka. In *Chittora et al. (2021)*, the authors utilized an SVM with wrapper feature selection methods, achieving an accuracy = 98.86, precision = 98.59, recall = 97.22, and F1-score = 97.90. Similarly, in *Qin et al. (2019)*, a hybrid model was employed, resulting in an impressive accuracy = 99.83, precision = 99.84, recall = 99.80, and F1-score = 99.86. In *Gunarathne, Perera & Kahandawaarachchi (2017)*, SVM was combined with filter FS methods and recorded accuracy = 98.5, precision = 98.6, recall = 98.5, and F1-score = 98.88. The work in *Polat, Danaei Mehr & Cetin (2017)* also utilized XGboost with filter FS methods, achieving accuracy = 99.10, precision = 99.5, recall = 99.33, and F1-score = 99.10. Additionally, *Dritsas & Trigka (2022)*, *Krishnamurthy et al. (2021)* applied RF without feature selection methods. As noticed in Table 8, no study in the literature used Confluent Cloud using Apache Kafka, and this is a great limitation of the literature study to provide real-time prediction systems. Our work recorded the highest performance based on accuracy, precision, recall, and F1-score. Also, it used GA as a feature selection method and developed a real-time system that integrates the proposed model with Confluent Cloud with Apache Kafka.

## CONCLUSION

This article delves into the role of ML in augmenting predictive capabilities within the healthcare domain, underscoring the pivotal role of stream processing platforms in crafting real-time streaming pipeline solutions. Our proposed system is structured into two phases: an offline model and an online prediction pipeline.

In the offline model phase, a stacking model is developed. GA and PSO were employed for feature selection, exploring a stacking model with the best features aided by XAI. The process involves several steps: preprocessing CKD benchmark data, utilizing GA and PSO for optimal feature selection, applying the stacking mode ML to selected features, and finally, employing XAI techniques to elucidate the stacking model with the best features. Standard evaluation metrics were used to assess model performance, including precision, recall, F1-score, and accuracy. Experimental results indicated that a stacking model with GA achieved the highest accuracy at 100.

Subsequently, the model was transformed for real-time execution. In the second phase, the model was applied to ingested data streams through a Python script, seamlessly integrating predictive analytics into real-time healthcare scenarios. Confluent Cloud was utilized, leveraging its capabilities to build real-time streaming systems using Apache Kafka and several APIs, including Producer API and Consumer API. A cluster and Kafka topics are generated for seamless connections between producers and consumers to Kafka topics. Streaming data pipelines were streamlined using Python scripts. Python producer script was created to generate streaming health attributes and push them into the Kafka topic. Python consumer script was created to consume health attributes from a Kafka topic and apply them to the model to predict CKD in real-time. Therefore, our framework contributes to advancing healthcare monitoring and improving decision-making processes in the healthcare field.

While the proposed framework shows the capability of combination stacking models, GA as feature selection, XAI, and big data streaming to enhance predictive capabilities in healthcare in real-time, several limitations need to be acknowledged. First, feature selection of GA may make high computational costs if it applies to large and complex datasets. Additionally, the initial setup and configuration of the streaming pipeline require a significant level of technical expertise, which could pose a barrier for institutions lacking sufficient IT support. Future work could investigate federated learning to allow models to be trained on distributed datasets stored in different hospitals or institutions without centralizing sensitive data, thus ensuring privacy while improving model robustness. In addition, the proposed framework can integrate with others big data platforms such as Apache Flink or Spark to support large streaming data. We can propose a real-time framework to support multi-modality data such as genetic information and imaging data to improve the diverse datasets.

### Funding

The authors received no funding for this work.

## Competing Interests

The authors declare that they have no competing interests.

## Author Contributions

- Abdulaziz AlMohimeed conceived and designed the experiments, performed the experiments, analyzed the data, performed the computation work, prepared figures and/or tables, authored or reviewed drafts of the article, and approved the final draft.

## Data Availability

The Chronic Kidney Disease Dataset is available at UCI machine learning repository: Rubini, L., Soundarapandian, P., & Eswaran, P. (2015). Chronic Kidney Disease [Dataset]. UCI Machine Learning Repository. https://doi.org/10.24432/C5G020.

The data is also available in the Supplemental File.

The disease prediction dataset and code is available at GitHub and Zenodo:

- https://github.com/Abdulazizal123/Disease_prediction.
- Abdulazizal123. (2025). Abdulazizal123/Disease_prediction: Disease prediction (v1.0). Zenodo. https://doi.org/10.5281/zenodo.15346624.

## Supplemental Information

Supplemental information for this article can be found online at http://dx.doi.org/10.7717/peerj-cs.2899#supplemental-information.

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
