# Peer review of "Cloud-based real-time enhancement for disease prediction using Confluent Cloud, Apache Kafka, feature optimization, and explainable artificial intelligence"

_PeerJ Computer Science, doi:10.7717/peerj-cs.2899_

## Round 0.1 · original submission · Major Revisions

Dear authors,

Thank you for the submission. The reviewers’ comments are now available. It is not suggested that your article be published in its current format. We do, however, advise you to revise the paper in light of the reviewers’ comments and concerns before resubmitting it. The followings should also be addressed:

1. Information about the datasets should be provided in the Abstract section.
2. Pay special attention to the usage of abbreviations. Spell out the full term at its first mention, indicate its abbreviation in parenthesis and use the abbreviation from then on.
3. The motivation and reason of using Genetic Algorithm and Particle Swarm Optimization method among many other metaheuristic algorithms for feature selection should be mentioned.
4. Equations should be used with correct equation number. Many of the equations are part of the related sentences. Attention is needed for correct sentence formation.
5. All of the values for the parameters of all algorithms should be given.
6. Encoding type or representation scheme and fitness function of Genetic Algorithm should be provided. How constraints for Particle Swarm Optimization (for example in decision variables intervals) are handled is not clear.
7. Reviewer1 has advised you to provide specific references. You are welcome to add them if you think they are relevant and useful . However, you are under no obligation to include them, and if you do not, it will not affect my decision.

Best wishes,

Reviewer 1 ·

Basic reporting

Please see the attachment.

Experimental design

Please see the attachment.

Validity of the findings

Please see the attachment.

Additional comments

Please see the attachment.

Annotated reviews are not available for download in order to protect the identity of reviewers who chose to remain anonymous.

·

Basic reporting

General Comments:
The article addresses an important subject, but several areas need improvement to strengthen its argumentation and overall readability. Key areas for revision include improving clarity, restructuring sections for better flow, enhancing technical explanations, and providing more evidence or data to support claims.
* * *
Abstract:
• Lack of Clarity: The abstract should succinctly summarize the problem, approach, key results, and significance of the study. Currently, it may lack focus.
• Suggestion: Rewrite the abstract to clearly state the research question or problem, the methodology used, the main findings, and their significance.
• Technical Jargon: Ensure that technical terms are either well explained or minimized in the abstract, as this section should be accessible to a broad audience.
* * *
Introduction:
• Problem Definition: The introduction does not clearly define the research problem or gap in the literature that the paper aims to address.
• Suggestion: Start with a clear statement of the research problem or gap. Clearly define why the topic is important and what specific issues you are addressing.
• Literature Review: There appears to be a lack of integration of existing literature. The introduction should include a review of relevant work to situate your contribution within the broader context.
• Suggestion: Add a subsection or integrate a concise literature review to show how your work builds on or diverges from previous studies.
• Objectives: The research objectives are not clearly stated.
• Suggestion: Clearly list the aims or objectives of the paper, and how they will contribute to solving the research problem.
* * *
Methodology:
• Lack of Detail: The methodology section is vague in several areas. The steps taken to conduct the study are not sufficiently explained.
• Suggestion: Provide more detailed descriptions of the methods used. If you used simulations, models, or experiments, give specifics about the tools, parameters, and techniques involved.
• Reproducibility: For research to be valuable, it must be reproducible. The methodology does not provide enough detail for other researchers to replicate the study.
• Suggestion: Include additional details such as the exact dataset used, any assumptions made, and parameters for models or simulations. This could involve adding a table, diagram, or flowchart to clarify the process.
* * *
Results:
• Lack of Depth: The results are either too brief or not well connected to the methodology. There should be a clear presentation of findings, supported by data.
• Suggestion: Expand the results section by presenting quantitative or qualitative data. Include tables, graphs, or charts to visualize the results. Ensure that the results are clearly linked back to the research objectives.
• Interpretation: Results are presented without much interpretation or discussion.
• Suggestion: After presenting the data, provide a detailed analysis and interpretation. Explain what the results mean in the context of the problem you are addressing.
* * *
Discussion:
• Weak Link to Research Problem: The discussion does not adequately connect the findings to the research problem and objectives outlined in the introduction.
• Suggestion: Revise the discussion to clearly link the results back to the original problem or hypothesis. Discuss how your findings address the research gap and what new knowledge they contribute to the field.
• Lack of Comparative Analysis: The paper lacks a comparison of the results with previous studies or methods.
• Suggestion: Include a comparison of your results with similar work from the literature. This strengthens the argument that your work is novel and significant.
* * *
Conclusion:
• Weak Conclusion: The conclusion is brief and does not effectively summarize the key findings or their implications.
• Suggestion: Expand the conclusion to summarize the key results, the contribution of the paper, and potential future work. Explain the broader impact of your findings and how they advance the field.
* * *
Technical and Formatting Issues:
1. Figures and Tables: Any figures or tables in the document should be clearly labeled and referenced in the text. Ensure that they are of high quality and contribute to understanding the results.
2. Suggestion: If there are missing or low-quality visuals, add or improve them. Make sure each visual element is referenced in the discussion.
3. Grammar and Style: The paper contains grammatical errors and awkward phrasing.
4. Suggestion: Go through the entire text and correct grammatical issues.
Recommendations for Revisions:
1. Reorganization of Content: Consider reorganizing the paper so that it follows a logical progression:
2. Introduction → Literature Review → Methodology → Results → Discussion → Conclusion.
3. Improved Argumentation: Strengthen the logic of your argumentation. Each section should naturally lead to the next, with clear transitions and linkages between ideas.
4. More Evidence: Wherever possible, provide more empirical evidence or data to support your claims. If relevant, include more case studies, experiments, or simulations that back up your conclusions.
5. Citations and References: Ensure all claims are properly cited. Check that your reference list is complete and follows the correct formatting style (e.g., APA, IEEE).
* * *
Recommendation: Major Revisions
By addressing these areas in your revision, the paper will become much stronger, clearer, and more compelling for readers. It will also be more aligned with academic standards for rigor and clarity.

Experimental design

As above

Validity of the findings

As above

Additional comments

As above

---

## Round 0.2 · Minor Revisions

Dear Authors,

One reviewer acceps your paper and other requests minor revision. It is recommended that the paper be revised and resubmitted in accordance with the Reviewer2's feedback.

Best wishes,

Reviewer 1 ·

Basic reporting

Yes

Experimental design

Sound good

Validity of the findings

Need to be improved.

Additional comments

Please review and check all reference styles according to the journal requirements. In addition, please read and cite these works (10.3390/diagnostics13111923; 10.1177/20552076231204748) in relevant sections.

·

Basic reporting

Decision: Minor Revision


The manuscript presents an innovative real-time system for early detection and treatment of Chronic Kidney Disease (CKD) by integrating machine learning models with big data streaming platforms. While the study is robust and comprehensive, there are a few minor issues that need to be addressed to ensure clarity, reproducibility, and consistency.

Experimental design

1. Abstract
Comment: The abstract lacks specific performance metrics (e.g., accuracy, precision, recall, F1-score).
2. Introduction
Comment: The introduction provides a good background but could benefit from emphasizing the novelty of the study.
3. Literature Review
Comment: The comparison with prior studies is comprehensive but lacks a summary table for clarity.
4. Methodology
Comment 1: The explanation of the stacking model and feature selection methods is clear but lacks visual aids for better understanding.
Comment 2: The parameters for GA and PSO are mentioned, but the rationale behind their selection is missing.
5. Results and Discussion
Comment 1: The results section does not provide statistical validation for the performance improvements achieved.
Comment 2: The discussion focuses heavily on numerical results but lacks insights into practical implications.

please highlight new comments in YELLOW color

Validity of the findings

provided above

Additional comments

Figures and Tables
Comment 1: Figures (e.g., ROC curves) lack detailed captions that explain their significance.
Suggestion: Enhance captions to include a brief interpretation of the results shown.
Comment 2: Table formatting is inconsistent, with some tables missing units or clear labeling.
Suggestion: Ensure all tables are uniformly formatted with proper headings and labels.
Writing and Formatting
Comment: There are minor grammatical errors and inconsistencies in terminology (e.g., "POS" vs. "PSO").
Suggestion: Proofread the manuscript carefully to correct errors and maintain consistency in terms and abbreviations.
Reproducibility
Comment: The manuscript does not provide access to the dataset or code for replicating the experiments.
Suggestion:
Include a link to the dataset or provide clear instructions on how to access it.
Share Python scripts or a GitHub repository for the offline and online prediction pipelines. Conclusion
Comment: The conclusion is well-written but does not propose concrete future research directions.
Suggestion: Suggest exploring multi-modal datasets, integrating other big data platforms, or applying the framework to additional diseases.
Additional Comments
Ethical Considerations: Briefly address the ethical implications of deploying such systems in healthcare, particularly regarding data privacy and security.
Broader Impact: Highlight how this framework could be adapted for other real-time prediction systems in healthcare or beyond.

---

## Round 0.3 · Minor Revisions

Dear Authors,

Thank you for submitting your article. Reviewer 2 has now commented on your article and suggests minor revisions. We do encourage you to address the concerns and criticisms of Reviewer 2 and resubmit your article once you have updated it accordingly.

Best wishes,

·

Basic reporting

The manuscript presents an innovative approach to real-time predictive healthcare analytics using IoT-based technologies, ensemble learning, and big data streaming platforms. Some minor revisions are required to enhance clarity, improve methodological robustness, and strengthen the evaluation framework.

Experimental design

Statistical Evaluation Enhancements

The manuscript primarily focuses on accuracy as a performance metric. To provide a more comprehensive evaluation, I recommend incorporating the following statistical methods for classification model comparison:

McNemar's Test: This test should be applied to compare the performance of paired classification models and assess whether their differences are statistically significant.

Kappa Statistic: This metric should be reported to measure inter-rater agreement, ensuring that the classification model is performing reliably across different instances.

Matthews Correlation Coefficient (MCC): MCC provides a more balanced measure of performance, particularly in cases where class imbalance exists. Including MCC will strengthen the analysis.

Validity of the findings

Experimental Setup and Validation

The pipeline's streaming process using Confluent Cloud and Apache Kafka is well described; however, it would be beneficial to:

Clearly outline the data sources and preprocessing steps used before feeding data into Kafka.

Provide more details on the real-time validation approach to ensure reproducibility.

Model Comparison and Performance Discussion

The results indicate that the stacking model with GA-based feature selection achieved 100% accuracy. While this is impressive, it is essential to discuss potential limitations such as:

The risk of overfitting due to an optimized feature set.

The generalizability of the model to new or unseen data.

A comparison of performance against baseline models (e.g., individual classifiers in the ensemble).

Additional comments

Clarity and Language Improvements

Several sections of the abstract and manuscript could benefit from improved readability and grammatical refinement. Specific suggestions include:

Replace "This paper uses a real simulation application to introduce a real-time system" with "This paper presents a real-time system for the early detection and treatment of Chronic Kidney Disease (CKD) using a real-world simulation application."

Clarify the role of explainable AI (XAI) by explicitly stating how it enhances the interpretability of the ensemble model.

Ensure consistency in terminology (e.g., "stacking model" vs. "stacking ensemble model").

---

## Round 0.4 · accepted · Accept

Dear Authors,

Thank you for addressing the reviewers' concerns and criticisms. Your manuscript now seems ready for publication. “is achieved the highest performance” should be “achieved the highest performance” in production step.

Best wishes,

·

Basic reporting

Minor Revision
Accept after addressing these comments
Grammar and Language Polishing:
Several areas in the manuscript would benefit from professional English proofreading.
Examples include:
“is achieved the highest performance”
should be “achieved the highest performance”.
“Hemoglobin has the second-highest positive contribution
Review for redundancy and flow.
Reference Addition:
Please cite the following relevant work
A Novel Study on Machine Learning Algorithm-Based Cardiovascular Disease Prediction

Experimental design

.

Validity of the findings

.

Additional comments

.